# Physics-informed Value Learner for Offline Goal-Conditioned Reinforcement Learning

**Vittorio Giammarino, Ruiqi Ni and Ahmed H. Qureshi**
Department of Computer Science
Purdue University
{vgiammar,ni117,ahqureshi}@purdue.edu

## Abstract

Offline Goal-Conditioned Reinforcement Learning (GCRL) holds great promise for domains such as autonomous navigation and locomotion, where collecting interactive data is costly and unsafe. However, it remains challenging in practice due to the need to learn from datasets with limited coverage of the state-action space and to generalize across long-horizon tasks. To improve on these challenges, we propose a *Physics-informed (Pi)* regularized loss for value learning, derived from the Eikonal Partial Differential Equation (PDE) and which induces a geometric inductive bias in the learned value function. Unlike generic gradient penalties that are primarily used to stabilize training, our formulation is grounded in continuous-time optimal control and encourages value functions to align with cost-to-go structures. The proposed regularizer is broadly compatible with temporal-difference-based value learning and can be integrated into existing Offline GCRL algorithms. When combined with Hierarchical Implicit Q-Learning (HIQL), the resulting method, Eikonal-regularized HIQL (Eik-HIQL), yields significant improvements in both performance and generalization, with pronounced gains in stitching regimes and large-scale navigation tasks. Code is available at link[1].

## 1 Introduction

In recent years, many of the most effective machine learning paradigms have capitalized on vast amounts of unlabeled or weakly labeled data. Similarly, in dynamic systems learning, Offline Goal-Conditioned Reinforcement Learning (GCRL) has emerged as a pivotal framework, enabling the use of large-scale, multitask datasets without requiring explicit reward annotations. Specifically, Offline RL [1, 2] leverages passively collected trajectories to learn control policies, offering great promise for applications such as autonomous navigation, locomotion, and manipulation, where interactive training is usually costly and unsafe. GCRL [3, 4] extends this capability by enabling learning across diverse datasets without explicit rewards. Despite its potential, Offline GCRL faces significant challenges, including accurate Goal-Conditioned Value Function (GCVF) estimation from limited data, policy extraction from imperfect value functions, and generalization to unseen state-goal pairs [5]. Among these issues, GCVF estimation remains the most fundamental, as improvements in this area can enhance both policy extraction and generalization, ultimately advancing the entire field of GCRL.

Physics-informed (Pi) inductive biases, defined as structural constraints grounded in physical laws such as symmetry, conservation principles, or consistency with Partial Differential Equations (PDEs), provide a promising direction for enhancing GCVF estimation in the offline setting. As demonstrated in prior work [6], Pi methods can introduce physically or geometrically meaningful structure into the learned value function, enhancing both sample efficiency and generalization. In Fig. 1, we illustrate a

---

[1]https://github.com/VittorioGiammarino/Eik-HIQL

39th Conference on Neural Information Processing Systems (NeurIPS 2025).

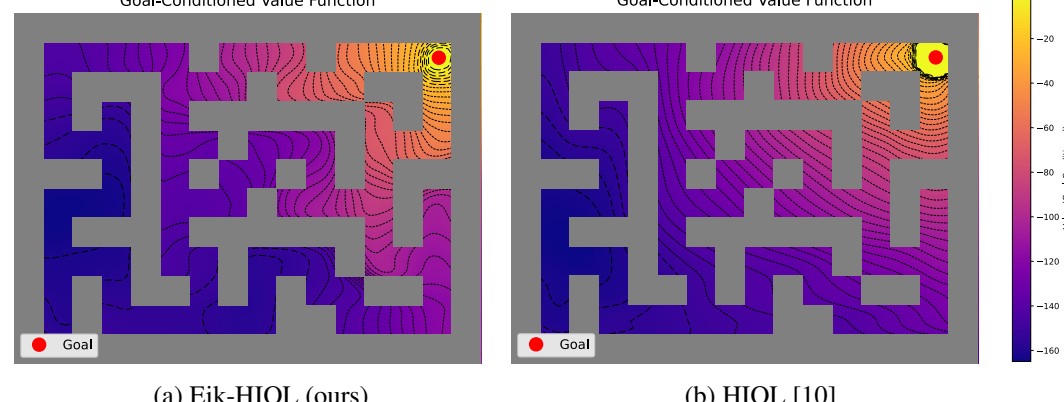

|  (a) Eik-HIQL (ours) | (b) HIQL [10] |

Figure 1: Contour plots of the GCVF for `antmaze-giant-navigate-v0` in [11], learned after 100,000 training steps by our Physics-informed algorithm **Eik-HIQL**, and the standard **HIQL**. The plots are generated by varying the agent's center of mass $x$-$y$ coordinates while keeping all other states fixed. Recall that the policy $\pi$ is trained to move the agent in the direction that maximizes the GCVF. The effects of the Eikonal regularizer are evident in Fig. 1a, where the contour plot closely follows the maze structure, in contrast to Fig. 1b, where the learned GCVF ignores the maze structure.

representative GCRL task in which an agent must navigate from various starting positions in a maze to a specified goal. Fig. 1b shows the contour plot of a GCVF learned by a non-Pi state-of-the-art (SOTA) algorithm. The resulting value function fails to robustly encode obstacle constraints, leading to suboptimal policies that often fail to reach the goal. These limitations motivate the use of Pi regularizers as a principled means to incorporate structural priors into value learning, and thus improve GCVF estimation in complex environments.

The primary contribution of this work is the introduction of an Eikonal regularizer for GCVF estimation in Offline GCRL tasks. Inspired by the Eikonal PDE [7], this regularizer imposes a distance-like cost-to-go structure on the learned GCVF, serving as an effective inductive bias during training. By enforcing this structure, the regularizer improves value estimation accuracy and promotes generalization to unseen states, while also reducing the number of required training steps compared to non-Pi approaches (see Fig. 1a). In contrast to Hamilton-Jacobi-Bellman (HJB) PDE-based methods [6], which require explicit system dynamics and often suffer from numerical instability [8, 9], our method is model-free and easy to implement. Empirically, it outperforms both HJB-regularized and unregularized baselines while adding only minimal computational overhead.

To validate the effectiveness of our Eikonal regularizer, we integrate it into the Hierarchical Implicit Q-Learning (HIQL) framework [10], a SOTA algorithm for Offline GCRL. We refer to this variant as Eik-HIQL. This choice is motivated by HIQL's strong baseline performance, making it an ideal candidate to highlight the benefits of our approach. Importantly, the Eikonal regularizer is broadly compatible with other temporal-difference-based algorithms, as we further demonstrate empirically in Appendix F. Our evaluation, conducted on the challenging OGbench benchmark [11], compares Eik-HIQL against Quasimetric RL (QRL) [12], Contrastive RL (CRL) [13], and the standard HIQL baseline. Eik-HIQL consistently outperforms or matches the baselines, achieving SOTA results in large-scale navigation and trajectory stitching scenarios. These gains underscore the utility of the Eikonal regularizer in enhancing GCVF estimation and overall Offline GCRL performance, with limited exceptions in tasks involving complex object interactions.

## 2 Related work

To the best of our knowledge, Pi regularization for value estimation has only recently been explored. Notably, Lien et al. [6] propose an Offline RL objective derived from the HJB equation in continuous-time optimal control [8, 9], aiming to enforce first-order derivative consistency within the critic network. In contrast, we introduce a simpler, model-free Pi regularizer for GCVF learning, based on the residual of the Eikonal PDE. We show that this regularizer induces a distance-like structure in the value function and integrates naturally into standard temporal-difference-based GCRL pipelines.

While gradient norm penalties, closely related to the Eikonal PDE residual, have been employed in generative modeling [14] and, more recently, in model-based RL to regularize Q-functions and mitigate overfitting [15], their application in GCRL remains, to our knowledge, unexplored. In contrast to these prior methods, which primarily aim to stabilize training, our regularizer is designed to inject a structural inductive bias into the GCVF, thereby improving sample efficiency and generalization. To the best of our knowledge, this work presents the first use of the Eikonal PDE as a regularization objective in value-based RL, and its first practical deployment in the Offline GCRL setting. More broadly, there has been a growing interest in incorporating physical priors and geometric, distance-like structures into RL algorithms, especially in model-based or Koopman-inspired frameworks [16, 17]. These approaches typically leverage the Koopman operator, which assumes access to (or approximations of) the underlying system dynamics, and may incorporate structural information such as reversibility or symmetry of the dynamics. In contrast, our method operates directly on the GCVF in a fully model-free setting.

In parallel, the use of non-Pi constraints for value learning has been extensively studied in Offline RL. Many approaches constrain learned policies to remain close to the behavior policy, either through explicit density modeling [18–20] or implicit divergence constraints [21, 22]. Others directly regularize the Q-function to assign low values to out-of-distribution actions and improve robustness [23, 24].

Our work is also closely related to the literature on GCRL [3, 4]. Eik-HIQL extends HIQL [10] by integrating the Eikonal regularizer into the GCVF estimation loss. HIQL itself combines hierarchical actors [25, 26] with Implicit Q-Learning [23]. Other GCRL methods include hindsight relabeling [27], contrastive representation learning [13], state-occupancy matching [28], and quasimetric RL [12]. Offline GCRL has also been studied through the lens of goal-conditioned supervised learning (GCSL) [29, 30], where goal-reaching policies are trained via conditional imitation or regression. Recent work has analyzed the out-of-distribution goal generalization problem [31] and addressed GCSL via self-supervised reward shaping [32] and occupancy-based score modeling [33].

Beyond RL, Pi losses and neural networks (NNs) have been widely applied to learn parameterizations of PDEs such as Burgers, Schrödinger, and Navier–Stokes equations [34, 35]. These methods leverage automatic differentiation to estimate derivatives with respect to NN inputs and solve high-dimensional PDEs. The Eikonal PDE, in particular, has been used in seismology [36] and motion planning [37, 38], where distance fields provide essential geometric structure. In this work, we extend the use of the Eikonal PDE to the Offline GCRL domain, demonstrating how its distance-preserving properties enhance GCVF estimation, especially in large environments and when data stitching is required.

## 3   Preliminaries

**Offline GCRL**   We model the decision process as a finite-horizon discounted Markov Decision Process (MDP) described by the tuple $(\mathcal{S}, \mathcal{G}, \mathcal{A}, \mathcal{T}, \mathcal{R}, \mathcal{P}_g, \rho_0, \gamma)$ where $\mathcal{S}$ is the set of states, $\mathcal{G}$ is the set of goals, $\mathcal{A}$ is the set of actions, $\mathcal{T} : \mathcal{S} \times \mathcal{A} \to \mathcal{P}(\mathcal{S})$ is the transition probability function where $\mathcal{P}(\mathcal{S})$ denotes the space of probability distributions over $\mathcal{S}$, $\mathcal{R} : \mathcal{S} \times \mathcal{G} \to \mathbb{R}$ is a goal-conditioned reward function defined $\mathcal{R}(s, g) = -1$ when $s \neq g$ and $\mathcal{R}(s, g) = 0$ otherwise, $\mathcal{P}_g \in \mathcal{P}(\mathcal{G})$ is the goal distribution, $\rho_0 \in \mathcal{P}(\mathcal{S})$ is the initial state distribution and $\gamma \in (0, 1]$ is the discount factor. In this work, we assume the goal space $\mathcal{G}$ to be equivalent to the state space, i.e., $\mathcal{G} = \mathcal{S}$, and the goal $g$ is sampled according to $\mathcal{P}_g$ ($g \sim \mathcal{P}_g$) at the beginning of each episode. The learning agent's objective is to maximize the expected sum of discounted rewards $\mathcal{R}(s_t, g)$ to successfully reach $g$. Formally, this objective is expressed as $J(\pi) = \mathbb{E}_{\tau_\pi(g)}[\sum_{t=0}^T \gamma^t \mathcal{R}(s_t, g)]$ where $\tau_\pi(g) = (g, s_0, a_0, s_1, a_1, \ldots, s_T)$ and $\pi : \mathcal{S} \times \mathcal{G} \to \mathcal{P}(\mathcal{A})$. Additionally, we define the GCVF induced by the policy $\pi$ as $V^\pi(s, g) = \mathbb{E}_{\tau_\pi(g)}[\sum_{t=0}^T \gamma^t \mathcal{R}(s_t, a_t, g)|S_0 = s, G = g]$ and the goal-conditioned state-action value function as $Q^\pi(s, a, g) = \mathbb{E}_{\tau_\pi(g)}[\sum_{t=0}^T \gamma^t \mathcal{R}(s_t, a_t, g)|S_0 = s, A_0 = a, G = g]$. In the offline setting, the learning agent must optimize $J(\pi)$ using only a static, offline dataset $\mathcal{D}$, which comprises trajectories of the form $\tau = (s_0, a_0, s_1, s_2, \ldots, s_T)$. Finally, note that we write $\pi_{\boldsymbol{\theta}}$ when a function is parameterized with parameters $\boldsymbol{\theta} \in \Theta \subset \mathbb{R}^k$.

**Hierarchical IQL**   Our algorithm, Eik-HIQL, extends the Offline GCRL algorithm HIQL, as introduced by Park et al. [10]. HIQL incorporates two key components: a GCVF estimation process that is robust to out-of-distribution actions and a hierarchical actor. The hierarchical actor comprises a high-level policy, $\pi^{hi}_{\boldsymbol{\theta}_{hi}} : \mathcal{S} \times \mathcal{G} \to \mathcal{S}$, which predicts subgoals, and a low-level policy, $\pi^{lo}_{\boldsymbol{\theta}_{lo}} : \mathcal{S} \times \mathcal{S} \to$

$\mathcal{P}(\mathcal{A})$, which generates actions to achieve those subgoals. Robustness in the GCVF estimation step is enabled by an action-free variant of implicit $Q$-learning [23], inspired by [39, 40]:

$$\mathcal{L}_V(\boldsymbol{\theta}_V) = \mathbb{E}_{(s,s')\sim\mathcal{D},g\sim\mathcal{P}_g}\Big[L_2^\iota\big(\mathcal{R}(s,g) + \gamma V_{\bar{\boldsymbol{\theta}}_V}(s',g) - V_{\boldsymbol{\theta}_V}(s,g)\big)\Big], \tag{1}$$

where $\bar{\boldsymbol{\theta}}_V$ denotes the parameters of the target GCVF network [41] and $L_2^\iota(\cdot)$ represents the expectile loss function with $\iota \in [0.5, 1]$: $L_2^\iota(x) = |\iota - \mathbb{1}(x < 0)|x^2$. In Eq. (1), the expectile regression induced by $L_2^\iota(\cdot)$ replaces the max operator in the Bellman equation [42] with the goal of avoiding queries of out-of-distributions actions. The ability to properly handle overestimated values for out-of-distribution actions is a critical challenge in Offline RL, since, unlike in online RL, erroneous estimates cannot be corrected through environment interactions. The estimated GCVF is subsequently used to train the hierarchical actor where $\pi_{\boldsymbol{\theta}_{hi}}^{hi}(s_{t+k}|s_t, g)$ and $\pi_{\boldsymbol{\theta}_{lo}}^{lo}(a|s_t, s_{t+k})$ aim to maximize $V_{\boldsymbol{\theta}_V}(s_{t+k}, g)$ and $V_{\boldsymbol{\theta}_V}(s_{t+1}, s_{t+k})$, respectively. It is shown in Park et al. [10] that this hierarchy, compared to the flat formulation using a single policy $\pi_{\boldsymbol{\theta}}(a|s_t, g)$, can better address low signal-to-noise ratios in the estimated GCVF.

**The Eikonal equation** The Eikonal equation is a non-linear first-order PDE that describes wave propagation in heterogeneous media [7]. It is expressed in its general form as:

$$||\nabla_s T(s,g)||^2 = \frac{1}{S(s)^2}, \tag{2}$$

where $||\cdot||$ denotes the Euclidean norm, $T : \mathcal{S} \times \mathcal{G} \to \mathbb{R}$ represents the travel-time through the medium from the state $s$ to a goal location $g$ and $\nabla_s T(s,g)$ is the partial derivative of the travel-time $T$ with respect to $s$. The function $S : \mathcal{S} \to \mathbb{R}$ defines the speed profile of the medium in the state location $s$. As described in [37], higher values for $S(s)$ lead to a low travel-time $T(s,g)$ from $s$ to $g$ and therefore to preferable paths $(s_0, s_1 \ldots, s_T)$ compared to those with lower $S(s)$. Consequently, the solution to the Eikonal PDE in (2), represented by the travel-time $T(s,g)$, reflects a cost-to-go structure. Minimizing $T(s,g)$ yields the shortest travel time from $s$ to $g$, as determined by the speed profile $S(s)$. In other words, $T(s,g)$ encodes a GCVF for a specific class of optimal control problems. We formally establish this connection in the next section. In our method, we propose a regularizer, inspired by the Eikonal PDE in (2), with the goal of providing an additional distance-like structure to the learned GCVF. Under smooth dynamics assumptions, we demonstrate that our regularizer significantly improves performance over the SOTA baselines while adding minimal complexity.

## 4 Physics-informed Eikonal regularizer

In this section, we first relate the Eikonal PDE in (2) to the HJB equation [8, 9] for continuous-time, undiscounted optimal control. This connection draws parallels to prior studies such as Lien et al. [6], offering additional motivation for our regularizer and insights into its empirical effectiveness. We then introduce the Eikonal-regularized loss for GCVF learning, which, when combined with a hierarchical actor, forms the core of our Eik-HIQL algorithm. Finally, we summarize the main components of Eik-HIQL.

**Optimal control perspective on the Eikonal PDE** We start our analysis by considering the following continuous-time dynamical system:

$$\dot{s}(t) = f(s(t), a(t)), \quad t \geq 0,$$

where $s(t)$ denotes the state of the system at time $t$, $\dot{s}(t)$ its derivative and $a(t)$ the control action. The function $f(\cdot)$ represents the system dynamics, determining how the state evolves in response to the control action. As common in the literature, we assume $f(\cdot)$ to be a Lipschitz continuous function [43]. Given the initial conditions $s(0) = s_0$, $a(0) = a_0$ and the goal $s(T) = g$, the undiscounted optimal control problem seeks to minimize

$$J = \int_0^T c(s(t), a(t))dt, \tag{3}$$

where $c(s(t), a(t))$ is the instantaneous cost function. The optimal value function $V(s,g)$ associated with (3) is defined as

$$V(s,g) = \inf_{a(\cdot)} \int_0^T c(s(t), a(t))dt,$$

and satisfies the principle of optimality

$$V(s, g) = \inf_{a \in \mathcal{A}} [c(s, a)\Delta t + V(s(t + \Delta t), g)], \tag{4}$$

where $\Delta t$ is a small time step. Note that, unless otherwise specified, throughout this section we keep the standard continuous-time optimal control theory notation, where $V(s, g)$ is used in place of $V^*(s, g)$ to denote the optimal value function [44].

Approximating $V(s(t + \Delta t), g)$ in (4) with its first-order Taylor expansion, $V(s(t + \Delta t), g) = V(s, g) + \nabla_s V(s, g)^\intercal f(s, a)\Delta t + O(\Delta t^2)$, and taking the limit $\Delta t \to 0$, we derive the following HJB equation:

$$\inf_{a \in \mathcal{A}} [c(s, a) + \nabla_s V(s, g)^\intercal f(s, a)] = 0, \tag{5}$$

which holds true at optimality. Refer to Appendix C for the step-by-step derivations. The HJB equation in (5) encodes the relationship between the system dynamics, the cost function, and the value function; where the left-hand side $H(s, g, \nabla_s V(s, g)) \equiv \inf_{a \in \mathcal{A}} [c(s, a) + \nabla_s V(s, g)^\intercal f(s, a)]$ is referred to as the Hamiltonian. The following proposition establishes the connection between the Hamiltonian and the Eikonal PDE in (2), highlighting their equivalence under specific conditions.

**Proposition 4.1.** *Given the Hamiltonian $H(s, g, \nabla_s V(s, g))$, the following inequality holds*

$$H(s, g, \nabla_s V(s, g)) \equiv \inf_{a \in \mathcal{A}} [c(s, a) + \nabla_s V(s, g)^\intercal f(s, a)] \leq c^*(s) + ||\nabla_s V(s, g)|| F^*(s), \tag{6}$$

*where $c^*(s) = \inf_{a \in \mathcal{A}} c(s, a)$ and $F^*(s) = \sup_{a \in \mathcal{A}} ||f(s, a)||$. In the special case in which $f(s, a) = a$, $||a|| = 1$ and $c(s, a)$ is constant over $||a|| = 1$ the Hamiltonian simplifies to*

$$H(s, g, \nabla_s V(s, g)) = c^*(s) - ||\nabla_s V(s, g)||. \tag{7}$$

*Proof.* The inequality in (6) follows from using the Cauchy-Schwarz inequality on the inner product $\nabla_s V(s, g)^\intercal f(s, a)$ in (5). Using the definitions $F^*(s) = \sup_{a \in \mathcal{A}} ||f(s, a)||$ and $c^*(s) = \inf_{a \in \mathcal{A}} c(s, a)$, we obtain the upper bound in (6). For the equality in (7), when $c(s, a)$ is constant over $||a|| = 1$, the inner product $\nabla_s V(s, g)^\intercal f(s, a)$ attains its minimal value when $a$ points in the direction opposite to $\nabla_s V(s, g)$. Specifically, this occurs when $a^* = \arg\inf_{||a||=1} \nabla_s V(s, g)^\intercal f(s, a) = -\nabla_s V(s, g)/||\nabla_s V(s, g)||$. Substituting $f(s, a) = a^*$ into the Hamiltonian in (6) and simplifying yields the result in (7). Refer to Appendix C for the full proof. $\square$

*Remark* 4.2 (Connection between HJB and Eikonal residuals). Proposition 4.1 shows that, even without assumptions on the dynamics, the Hamiltonian $H(s, g, \nabla_s V(s, g))$ is upper-bounded by an Eikonal-like residual, where the ratio $F^*(s)/c^*(s)$ defines a local speed profile $S(s)$ as in (2). In the special case of isotropic dynamics with $f(s, a) = a$, $||a|| = 1$, and constant cost, the Hamiltonian reduces exactly to the Eikonal PDE with $S(s) = 1/c^*(s)$. Thus, while the HJB PDE formalizes cost-to-go under known dynamics, the Eikonal PDE captures a related spatial structure through $S(s)$, making it a natural approximation when dynamics are unknown.

*Remark* 4.3 (Why the Eikonal residual helps in Offline GCRL). Temporal-difference learning (e.g., (1)) is known to converge to the optimal GCVF $V^*$ under ideal conditions: namely, on-policy data and an infinite sample budget [42]. However, these assumptions are often violated in the offline setting, where biased datasets and long-horizon tasks exacerbate extrapolation error and limit generalization [10]. In this context, the Eikonal residual in Eq. (7) can play a crucial role by introducing a geometric inductive bias that encourages the learned value function to behave like a distance field, through the constraint $||\nabla_s V_{\boldsymbol{\theta}}(s, g)|| \approx c(s)$. This regularization is particularly effective when the true $V^*$ is Lipschitz continuous, i.e., in environments where the dynamics do not induce sharp discontinuities [45]. Under such conditions, the Eikonal residual shapes the gradient norm of the learned GCVF $V_{\boldsymbol{\theta}}$ to match the local structure of $V^*$, up to a scaling factor. This effect is supported by Rademacher's theorem [46], which guarantees that Lipschitz continuous functions are differentiable almost everywhere, with $||\nabla_s V^*(s, g)|| \leq L$, where $L$ is the minimal Lipschitz constant. As a result, when $V^*$ is Lipschitz continuous, the Eikonal residual in Eq. (7) provides a principled inductive bias that, as we empirically demonstrate, improves both sample efficiency and generalization, particularly in long-horizon tasks with smooth dynamics.

Furthermore, in practice, the Eikonal residual offers a tractable alternative to HJB regularization in model-free settings where the dynamics $f(s, a)$ are unknown. Prior work [6] approximates the HJB

term using finite differences, i.e., $f(s, a) \approx s' - s$, but our experiments show no clear advantage over the simpler Eikonal residual regularization. Moreover, note that, while the HJB equation in Eq. (5) holds only at optimality, the inequality in Proposition 4.1 remains valid throughout training, providing consistent structural guidance even when $V$ is suboptimal.

**Eikonal-regularized implicit $V$-learning** Based on the upper-bound in Proposition 4.1 and the discussion in Remark 4.3, we propose the following Eikonal-regularized implicit $V$-learning loss for GCVF estimation:

$$\mathcal{L}_V(\boldsymbol{\theta}_V) = \mathbb{E}_{(s,s')\sim\mathcal{D}, g\sim\mathcal{P}_g}\Big[L_2^\iota\big(\mathcal{R}(s,g) + \gamma V_{\bar{\boldsymbol{\theta}}_V}(s',g) - V_{\boldsymbol{\theta}_V}(s,g)\big) \tag{8}$$

$$+ \big(||\nabla_s V_{\boldsymbol{\theta}_V}(s,g)|| \cdot S(s) - 1\big)^2\Big], \tag{9}$$

where the term in (8) corresponds to the expectile regression in (1) (cf. [10]) and (9) is our Eikonal regularizer. In (9), $||\nabla_s V_{\boldsymbol{\theta}_V}(s,g)||$ is the Euclidean norm of the GCVF gradient with respect to its input $s$ and $S(s)$ is a pre-specified speed profile that maps states to scalar values (see Preliminaries in Section 3). The term $\nabla_s V_{\boldsymbol{\theta}_V}(s,g)$ is computed via automatic differentiation, following standard approaches in PiNNs (see Algorithm 2 in Appendix D). The speed profile $S(s)$ is designed such that it encapsulates additional biases or task-specific information into the Eikonal regularizer [37]. We demonstrate in our experiments that the simple choice of $S(s) = 1$ works best in practice; however, we believe that more interesting designs might further improve collision avoidance in cluttered environments and consequently enhance both safety and performance. We defer this direction on Pi Safe GCRL to future work. Finally, note that in (9), with respect to the upper bound in (6), we set $c^*(s) = -1$ and opt to multiply $S(s)$ with $||\nabla_s V_{\boldsymbol{\theta}_V}(s,g)||$ rather than using $c^*(s) = -1/S(s)$ as in (2). This in order to ensure numerical stability.

As discussed in Remark 4.3, the effectiveness of the regularization in Eq. (9) stems from the geometric structure of the Eikonal PDE, whose solution defines a signed distance field [36, 37]. In our final formulation, we uniformly set the speed profile to $S(s) = 1$, corresponding to the constraint $||\nabla_s V_{\boldsymbol{\theta}_V}(s,g)|| \approx 1$ across the feasible state space. By Rademacher's theorem [46], this enforces 1-Lipschitz continuity almost everywhere, encouraging smoother and more generalizable value estimates even under limited offline data coverage. Although the true optimal value function $V^*$ may not be exactly 1-Lipschitz over the entire feasible space, we show in our experiments (Section 5) that this regularization remains effective in practice. Note that using a different constant $S(s) > 0$ everywhere in the feasible space would simply uniformly rescale the value function without affecting its shape. Consequently, since policy gradients are invariant to such rescaling [42, 47, 48], this design choice has no impact on the induced policy.

**Eikonal-regularized HIQL** In the following, we provide a brief summary of our algorithm, Eik-HIQL. During training, Eik-HIQL performs an Eikonal-regularized value estimation step, where the loss in (8)-(9) is minimized to learn a GCVF. This is followed by a policy extraction step, in which the hierarchical actor introduced in Park et al. [10] (see Preliminaries in Section 3) is trained based on the estimated GCVF. The full pseudo-code for Eik-HIQL as well as a JAX [49] implementation showing how to compute the gradient $\nabla_s V_{\boldsymbol{\theta}_V}$ in (9) are provided in Appendix D, respectively Algorithm 1 and Algorithm 2. For the full implementation of Eik-HIQL refer to our GitHub repository.

## 5 Experiments

In this section, we analyze the effects of the Eikonal regularizer in (9) on the GCVF estimation problem. Specifically, we will perform an ablation over different designs of speed profiles $S(s)$, compare the Eikonal regularizer with an HJB regularizer, and analyze value functions learned with and without our Eikonal term. Then, we compare the performance obtained by our Eikonal-regularized algorithm, Eik-HIQL, against the SOTA algorithms for Offline GCRL. Finally, we will also present the limitations of our approach. The experiments in this section are conducted on the environments in Fig. 2. A table summarizing the most relevant hyperparameter values is provided in Appendix D.

**Speed profiles and HJB regularizer comparison** Recall that our algorithm, *Eik-HIQL*, estimates the GCVF using the loss defined in Eqs. (8)-(9), where the speed profile $S(s)$ encodes task-specific structure into the Eikonal regularizer. We perform an ablation over different choices of $S(s)$ and

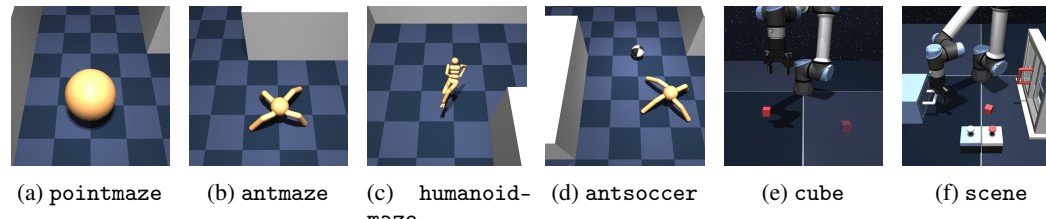

(a) `pointmaze`   (b) `antmaze`   (c) `humanoid-` (d) `antsoccer`   (e) `cube`   (f) `scene`
`maze`

Figure 2: Environments from OGbench [11] used in our experiments. These include a variety of goal-conditioned tasks spanning navigation and locomotion (e.g., `pointmaze`, `antmaze`, `humanoidmaze`), contact-rich locomotion (`antsoccer`), and contact-rich manipulation (`cube`, `scene`). The environments differ significantly in dynamics complexity, dimensionality, and task structure, providing a comprehensive testbed for evaluating Offline GCRL algorithms.

Table 1: Summary of the speed profiles ablation. All agents are trained for **100,000 training steps** using 10 seeds. We report the mean and standard deviation across seeds for the best evaluation achieved during training. For each seed, evaluations are conducted over 5 different random goals, as designed in Park et al. [11], with the learned policy tested for 50 episodes per goal. Results within $95\%$ of the best value are written in **bold**.

| Environment | Dataset Type | Maze Dimension | Eik-HIQL | Eik-HIQL Exp (10) | Eik-HIQL Lin (11) | HJB-HIQL (12) |
|---|---|---|---|---|---|---|
| pointmaze | navigate | medium | $\mathbf{94 \pm 4}$ | $\mathbf{95 \pm 2}$ | $\mathbf{93 \pm 4}$ | $90 \pm 6$ |
| | | large | $\mathbf{83 \pm 9}$ | $61 \pm 8$ | $60 \pm 5$ | $53 \pm 9$ |
| | | giant | $\mathbf{79 \pm 13}$ | $38 \pm 15$ | $42 \pm 12$ | $9 \pm 8$ |
| | | teleport | $\mathbf{47 \pm 10}$ | $39 \pm 7$ | $40 \pm 6$ | $18 \pm 5$ |
| | stitch | medium | $\mathbf{97 \pm 2}$ | $90 \pm 9$ | $85 \pm 10$ | $64 \pm 13$ |
| | | large | $\mathbf{73 \pm 6}$ | $33 \pm 17$ | $41 \pm 7$ | $6 \pm 7$ |
| | | giant | $\mathbf{22 \pm 10}$ | $5 \pm 8$ | $2 \pm 3$ | $0 \pm 0$ |
| | | teleport | $\mathbf{44 \pm 9}$ | $\mathbf{43 \pm 5}$ | $38 \pm 7$ | $18 \pm 8$ |

find the simple constant setting $S(s) = 1$ to be particularly effective. It outperforms more complex alternatives that require explicit knowledge of obstacle coordinates, which may not be available in practice. Specifically, we compare $S(s) = 1$ against the following two speed profiles:

$$S_{\text{exp}}(s) = S_{\text{min}} + (1 - S_{\text{min}}) \exp\left(-\lambda \frac{d_{\text{max}} - d(s)}{d_{\text{max}} - d_{\text{min}}}\right), \tag{10}$$

$$S_{\text{lin}}(s) = \text{clip}\left(\frac{d(s)}{d_{\text{max}}}, \frac{d_{\text{min}}}{d_{\text{max}}}, 1.0\right), \tag{11}$$

where $\lambda$ is a decay rate, $S_{\text{min}}$ represents the minimum tolerable speed, $d_{\text{min}}$ and $d_{\text{max}}$ are respectively the minimum and maximum tolerable distances of a state $s$ from its nearest obstacle and $d(s)$ is a function $d : \mathcal{S} \to \mathbb{R}$ describing the Euclidean distance of the state $s$ from its nearest obstacle (see Appendix B for more information). We refer to $S_{\text{exp}}(\cdot)$ in (10) as the exponential speed profile (*Eik-HIQL Exp* in Table 1) and to $S_{\text{lin}}(\cdot)$ in (11), originally introduced in [37], as the linear speed profile (*Eik-HIQL Lin* in Table 1). Both (10) and (11) assign high speed values to states $s$ far from obstacles, and low speed values to states near obstacles. This encourages the agent to avoid obstacles, as proximity to them results in longer travel-time from $s$ to $g$. Moreover, due to the use of the $\exp(\cdot)$ function, when compared to $S_{\text{lin}}(\cdot)$; $S_{\text{exp}}(\cdot)$ ensures a smoother decay of the speed value as the distance from the obstacles decreases [50]. Finally, note that, in both these speed profiles, computing $d(s)$ requires knowledge of the obstacles' coordinates which represents a strong assumption in some settings. In addition to this ablation, we also compare our Eikonal regularization term in (9) with an HJB regularizer derived from the HJB PDE in (5):

$$\mathcal{L}_V^{\text{HJB}}(\boldsymbol{\theta}_V) = \mathbb{E}_{(s,s') \sim \mathcal{D}, g \sim \mathcal{P}_g}\left[\left(\nabla_s V_{\boldsymbol{\theta}_V}(s, g)^{\mathsf{T}}(s' - s) - 1\right)^2\right]. \tag{12}$$

In (12), the dynamics $f(s, a)$, originally required by the HJB PDE in (5), is replaced by a finite difference term $(s' - s)$ as proposed by Lien et al. [6]. The term in (12) is then used in place of the Eikonal regularizer in (9) for GCVF estimation in Eik-HIQL. We refer to this approach as *HJB-HIQL* in Table 1. The results for these experiments are summarized in Table 1 and all the learning curves are available in Appendix E.

Two main observations explain the superior performance of the simple choice $S(s) = 1$. First, in Offline GCRL, sampled trajectories are already constrained to the feasible space, and there is no explicit penalty for colliding with obstacles during learning. As such, complex speed profiles incorporating obstacle proximity provide limited additional benefit over the uniform case. Second, the simplicity of $S(s) = 1$ enables more efficient and stable learning of the GCVF, whereas profiles requiring privileged information (e.g., obstacle maps) may introduce unnecessary complexity. Our choice of $S(s) = 1$ also aligns with standard practice in the literature, which commonly adopts uniform speed profiles to model wavefront propagation in the feasible state space [37, 51]. This also ensures a fair comparison with the Offline GCRL baselines, as none of them leverage privileged information. Furthermore, we provide visualizations for the learned GCVF in Appendix B, where contour plots for `pointmaze-giant-stitch-v0` illustrate that *Eik-HIQL* with $S(s) = 1$ learns a more structured and accurate value function, closely aligning with the maze layout. In contrast, *Eik-HIQL Exp* and *Eik-HIQL Lin* display artifacts even near the goal, while *Eik-HIQL HJB* fails to recover the maze geometry altogether. Based on this empirical evidence, and to ensure a fair comparison with baselines, we adopt the constant speed profile $S(s) = 1$ in all subsequent experiments. Nonetheless, we highlight that in the context of Safe GCRL, where value functions must encode both safety and task performance, investigating how different choices of $S(s)$ influence the shape and behavior of the GCVF represents an interesting direction for future work.

**Eik-HIQL vs HIQL** We compare Eik-HIQL with its non-regularized counterpart, HIQL [10], to isolate the effect of the Eikonal regularizer. This comparison is conducted under tightly controlled conditions: both methods use identical network architectures, hyperparameters, and training pipelines, ensuring that the only difference lies in the presence of the regularizer. As shown in Table 2, Eik-HIQL consistently outperforms HIQL on navigation tasks, where the value function is typically Lipschitz continuous. The benefits are particularly pronounced in large mazes and stitching regimes, with improvements exceeding 100% in 7 out of 31 evaluated tasks.

Given the magnitude of these gains relative to the standard deviations across 10 random seeds, the improvements are both practically meaningful and statistically significant. Full learning curves for these experiments are provided in Appendix E. These results highlight the robustness of the Eikonal regularizer in smooth environments and demonstrate its ability to enhance generalization in settings where HIQL struggles to scale. As illustrated in Fig. 1 for the

Table 2: Complete comparison between Eik-HiQRL and the Offline GCRL baselines. Agents are trained for **100,000 steps** on `pointmaze` tasks and **1 million steps** on the remaining tasks, each using 10 seeds. The evaluation follows the methodology described in Table 1. We report the mean and standard deviation across seeds for the best evaluation achieved during training. Results within 95% of the best value are written in **bold**, and rows are highlighted when the Eikonal regularizer improves performance by 100% or more compared to the non-regularized HIQL performance.

| Environment | Dataset | Dim | Eik-HIQL | HIQL | QRL | CRL |
|---|---|---|---|---|---|---|
| pointmaze | navigate | medium | **93 ± 5** | **92 ± 2** | 83 ± 3 | 54 ± 19 |
| | | large | 83 ± 9 | 49 ± 13 | **90 ± 5** | 56 ± 9 |
| | | giant | **79 ± 13** | 7 ± 8 | 72 ± 7 | 37 ± 17 |
| | | teleport | 47 ± 10 | 29 ± 7 | 34 ± 7 | **50 ± 5** |
| | stitch | medium | **96 ± 3** | 76 ± 8 | 80 ± 10 | 3 ± 5 |
| | | large | 73 ± 6 | 19 ± 7 | **85 ± 11** | 4 ± 6 |
| | | giant | 22 ± 10 | 1 ± 4 | **56 ± 9** | 0 ± 0 |
| | | teleport | 43 ± 9 | 38 ± 5 | **42 ± 6** | 12 ± 6 |
| antmaze | navigate | medium | **95 ± 1** | **96 ± 1** | 87 ± 5 | **94 ± 2** |
| | | large | **86 ± 2** | **90 ± 6** | 80 ± 5 | **86 ± 3** |
| | | giant | 67 ± 5 | **69 ± 3** | 14 ± 6 | 18 ± 4 |
| | | teleport | 52 ± 4 | 43 ± 3 | 39 ± 4 | **55 ± 4** |
| | stitch | medium | **94 ± 2** | **95 ± 14** | 68 ± 6 | 54 ± 8 |
| | | large | **84 ± 3** | 74 ± 6 | 24 ± 5 | 13 ± 4 |
| | | giant | **48 ± 11** | 3 ± 3 | 2 ± 2 | 0 ± 0 |
| | | teleport | **47 ± 2** | 35 ± 3 | 29 ± 6 | 34 ± 4 |
| | explore | medium | **43 ± 15** | 33 ± 15 | 5 ± 4 | 4 ± 2 |
| | | large | **13 ± 1** | 6 ± 7 | 0 ± 0 | 0 ± 0 |
| | | teleport | 15 ± 10 | **45 ± 5** | 2 ± 2 | 22 ± 5 |
| humanoidmaze | navigate | medium | 86 ± 2 | **90 ± 3** | 22 ± 2 | 61 ± 4 |
| | | large | **64 ± 7** | 50 ± 4 | 7 ± 3 | 22 ± 9 |
| | | giant | **68 ± 5** | 18 ± 5 | 1 ± 1 | 4 ± 2 |
| | stitch | medium | 79 ± 2 | **88 ± 3** | 22 ± 4 | 40 ± 7 |
| | | large | **29 ± 7** | **28 ± 2** | 3 ± 1 | 4 ± 2 |
| | | giant | **19 ± 5** | 3 ± 1 | 0 ± 0 | 0 ± 0 |
| antsoccer | navigate | arena | 19 ± 2 | **60 ± 4** | 10 ± 3 | 24 ± 2 |
| | | medium | 3 ± 2 | **13 ± 3** | 2 ± 2 | 4 ± 2 |
| | stitch | arena | 2 ± 0 | **17 ± 3** | 2 ± 1 | 1 ± 1 |
| | | medium | 1 ± 0 | **5 ± 1** | 0 ± 0 | 0 ± 0 |
| manipulation | cube-single-play | | 25 ± 1 | **31 ± 4** | 11 ± 3 | **32 ± 2** |
| | scene-play | | **52 ± 7** | **52 ± 3** | 8 ± 2 | 35 ± 6 |

`antmaze-navigate-giant-v0` task, Eik-HIQL produces a GCVF that reflects the underlying maze geometry, whereas HIQL fails to capture this structure, leading to poor goal-directed performance.

By contrast, in interactive, contact-rich domains such as `antsoccer` and `manipulation`, where the dynamics (and consequently the value function) exhibit discontinuities, Eik-HIQL offers limited advantage. We defer a detailed discussion of these cases to the Limitations section below.

**Offline GCRL**  We extend our comparison to state-of-the-art Offline GCRL algorithms, including QRL [12] and CRL [13], alongside HIQL. Results across all 31 tasks are summarized in Table 2, with full learning curves in Appendix E. Eik-HIQL consistently outperforms all baselines in the most challenging settings, most notably in the `antmaze-stitch` tasks, which combine complex locomotion with, composite datasets, and in `humanoidmaze`, where high-dimensional states and unstable control amplify learning difficulty.

QRL performs well in simpler domains such as `pointmaze`, where its quasimetric structure aids goal-conditioned estimation. CRL is competitive on several `navigate` variants, but struggles in high-dimensional and stitched tasks such as `humanoidmaze`. In contrast, Eik-HIQL demonstrates strong generalization and planning performance in both long-horizon and large-scale environments.

In `antsoccer` and `manipulation`, however, Eik-HIQL performs comparably to the baselines. As mentioned, these domains involve discontinuous or contact-rich dynamics, and we do not observe consistent benefits from the Eikonal regularizer due to the fact that the imposed gradient condition does not hold globally at optimality throughout the entire state space.

**Limitations**  We conclude by discussing some limitations of our approach, particularly in contact-rich tasks that involve interactions with external objects. In the `antsoccer` domain, for instance, an ant agent must not only navigate but also interact with a soccer ball to reach a specified goal. Similarly, in manipulation tasks such as `cube-single-play`, the agent must coordinate precise contact interactions with objects in the scene. These tasks introduce hybrid, non-Lipschitz dynamics due to discontinuities in contact states, often modeled as categorical variables (e.g., in-contact vs. free motion), which pose a challenge for regularizers grounded in smoothness and Lipschitz continuity assumptions [43], such as our Eikonal term. As visu-

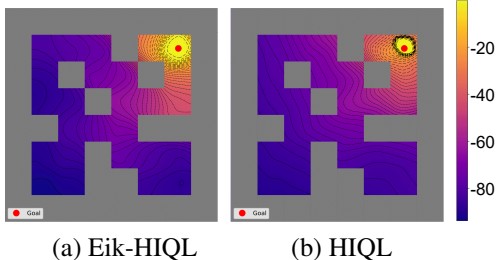

(a) Eik-HIQL         (b) HIQL

Figure 3: Countour plots of the GCVF on `antsoccer-medium-navigate-v0` [11], learned after 1 million steps by Eik-HIQL and HIQL respectively. These plots are generated following the same methodology in Fig. 1.

alized in Fig. 3 for the `antsoccer` experiments, Eik-HIQL learns a GCVF that aligns well with the environment geometry, particularly for the navigational components. However, this structured value function does not consistently yield better performance (cf. Table 2) as these tasks also require complex interactions with objects in the environment. Similar trends are observed in manipulation tasks, where Eik-HIQL performs comparably to the baselines but does not show marked gains. These experiments are included for completeness and highlight that, while Eik-HIQL excels in navigation-dominated domains, additional mechanisms, such as task-adaptive speed profiles or representation learning tailored to contact dynamics, may be necessary to extend its benefits to interactive, contact-rich environments. We leave this direction for future work.

## 6  Conclusion

We introduced Eik-HIQL, a novel approach to Offline GCRL that integrates an Eikonal PDE-based regularizer for GCVF estimation. Our analysis demonstrated that the Eikonal regularizer effectively introduces a useful distance-like inductive bias, which promotes consistent gradient magnitudes and improves value estimation in high-dimensional spaces (cf. Fig. 1). This property mitigates irregularities from the limited coverage of offline datasets, resulting in robust, globally consistent GCVFs that accurately capture the underlying structure of the environment. Consequently, Eik-HIQL outperformed SOTA baselines across diverse tasks, excelling in complex scenarios such as large-scale mazes and trajectory stitching, where traditional methods often fail to generalize.

However, our experiments also highlighted limitations in interactive tasks. We discuss how our Eikonal regularizer induces excessive smoothness in the learned GCVFs which does hinder perfor-

mance in interactive tasks where non-smoothness, or at least non-global smoothness, is required. These findings suggest that, while the Eikonal regularizer significantly enhances navigation tasks, future work should incorporate mechanisms that better capture task-specific dynamics, such as object interaction, to improve applicability across diverse domains.

Overall, this work underscores the potential of Pi methods to address fundamental challenges in Offline GCRL. By enhancing scalability and generalization of value estimation, the Eikonal regularizer provides a foundation for leveraging domain knowledge in RL. Future research could expand on this foundation by exploring multi-agents settings and integrating task-specific biases for interactive environments.

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

# A  Impact Statement

This paper aims to advance the field of machine learning for autonomous decision-making and control in robotic systems. We achieved this goal through the development of physics-informed methods for offline goal-conditioned reinforcement learning. While our work has potential societal implications, we do not identify any that require specific emphasis here.

# B  Speed profiles and HJB comparison: additional details and contour plots

In this section, we provide additional details related to the *Speed Profiles and HJB Regularizer Comparison* paragraph in Section 5. Specifically, we illustrate how the distance function $d(s)$, used to define the speed profiles in (10) and (11), is computed in Fig. 4.

Additionally, Fig. 5 presents contour plots of the GCVFs learned by the algorithms evaluated in Table 1. These visualizations provide qualitative support for the quantitative results in Table 1, showing that higher returns tend to correlate with smoother, artifact-free value functions that more closely follow the structure of the underlying maze.

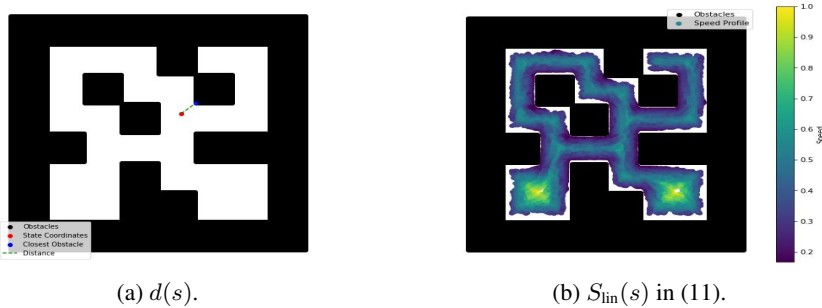

(a) $d(s)$.                                              (b) $S_{\text{lin}}(s)$ in (11).

Figure 4: Fig. 4a illustrates the computation of the distance function $d(s)$ used in (10) and (11). Let the state be represented by its spatial coordinates $s = (x, y) \in \mathbb{R}^2$, and let $\mathcal{O} = \{o_1, \ldots, o_M\}$ denote the set of obstacle coordinates in the maze. We define $d(s) = \min_{o \in \mathcal{O}} \|s - o\|_2$, i.e., the Euclidean distance from $s$ to the nearest obstacle. Fig. 4b reports the resulting speed profile obtained using $S_{\text{lin}}(s)$ in (11) for the `pointmaze-medium-navigate-v0` dataset.

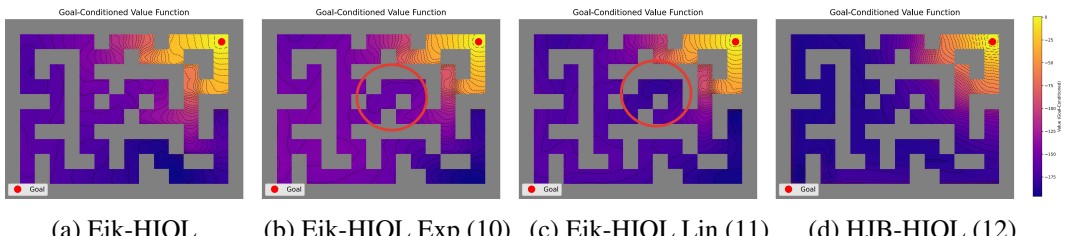

(a) Eik-HIQL        (b) Eik-HIQL Exp (10)   (c) Eik-HIQL Lin (11)    (d) HJB-HIQL (12)

Figure 5: Contour plots of the GCVF on `pointmaze-giant-stitch-v0` [11], learned after 100,000 training steps by the algorithms in Table 1. We observe that Eik-HIQL with constant speed profile $S(s) = 1$ provides the most accurate GCVF estimation, resulting in the highest score for this task. In contrast, the contour plot for HJB-HIQL in (d) fails to effectively capture the maze layout. Furthermore, in (b) and (c) we annotate, with red circles, examples of artifacts arising in Eik-HIQL Exp and Eik-HIQL Lin, respectively.

# C  HJB PDE step-by-step derivations and Proof of Proposition 4.1

In the following we provide the step-by-step derivations for the HJB PDE in (5) and the full proof for Proposition 4.1.

## C.1 HJB PDE derivations

The optimal value function $V(s, g)$ associated with the undiscounted optimal control problem in (3), satisfies the following principle of optimality

$$V(s, g) = \inf_{a \in \mathcal{A}} [c(s, a)\Delta t + V(s(t + \Delta t), g)], \tag{13}$$

where $\Delta t$ is a small time step. By substituting the Taylor expansion

$$V(s(t + \Delta t), g) = V(s, g) + \nabla_s V(s, g)^\top f(s, a)\Delta t + O(\Delta t^2)$$

into (13), we obtain

$$V(s, g) = \inf_{a \in \mathcal{A}} [c(s, a)\Delta t + V(s, g) + \nabla_s V(s, g)^\top f(s, a)\Delta t + O(\Delta t^2)].$$

Subtracting $V(s, g)$ on both sides and dividing by $\Delta t$ gives:

$$0 = \inf_{a \in \mathcal{A}} [c(s, a) + \nabla_s V(s, g)^\top f(s, a) + O(\Delta t)].$$

Taking the limit $\Delta t \to 0$, we recover Eq. (5):

$$\inf_{a \in \mathcal{A}} [c(s, a) + \nabla_s V(s, g)^\top f(s, a)] = 0.$$

## C.2 Proof of Proposition 4.1

*Proof.* Consider the HJB PDE in (5). By applying the Cauchy-Schwarz inequality to the argument of the minimization we obtain

$$c(s, a) + \nabla_s V(s, g)^\top f(s, a) \le c(s, a) + \|\nabla_s V(s, g)\|\|f(s, a)\|.$$

Then, by defining

$$F^*(s) = \sup_{a \in \mathcal{A}} \|f(s, a)\|,$$

we can further upper bound the right-hand-side and obtain:

$$c(s, a) + \nabla_s V(s, g)^\top f(s, a) \le c(s, a) + \|\nabla_s V(s, g)\| F^*(s).$$

Finally, applying the infimum over $a \in \mathcal{A}$ on both sides yields:

$$\inf_{a \in \mathcal{A}} [c(s, a) + \nabla_s V(s, g)^\top f(s, a)] \le \inf_{a \in \mathcal{A}} [c(s, a) + \|\nabla_s V(s, g)\| F^*(s)],$$

where the result in Eq. (6) is obtained by defining $c^*(s) = \inf_{a \in \mathcal{A}} c(s, a)$.

For the equality in (7), note that for the isotropic dynamics $f(s, a) = a$ and given $c(s, a)$ constant over $\|a\| = 1$, the inner product $\nabla_s V(s, g)^\top f(s, a)$ attains its minimal value when $a$ points in the direction opposite to $\nabla_s V(s, g)$. Specifically, this occurs when

$$a^* = \arg \inf_{\|a\|=1} \nabla_s V(s, g)^\top a = -\frac{\nabla_s V(s, g)}{\|\nabla_s V(s, g)\|}.$$

Substituting $f(s, a) = a^*$ into

$$H(s, g, \nabla_s V(s, g)) = \inf_{a \in \mathcal{A}} [c(s, a) + \nabla_s V(s, g)^\top f(s, a)]$$

and simplifying yields

$$H(s, g, \nabla_s V(s, g)) = c^*(s) - \|\nabla_s V(s, g)\|.$$

$\square$

# D  Psuedocode and Hyperparameters

During training, Eik-HIQL performs an Eikonal-regularized value estimation step followed by a hierarchical policy extraction step. During the value estimation step, the following loss, as provided in (8)-(9), is minimized:

$$
\mathcal{L}_V(\boldsymbol{\theta}_V) = \mathbb{E}_{(s_t, s_{t+1}) \sim \mathcal{D}, g \sim \mathcal{P}_g} \Big[ L_2^\iota \big( \mathcal{R}(s_t, g) + \gamma V_{\bar{\boldsymbol{\theta}}_V}(s_{t+1}, g) - V_{\boldsymbol{\theta}_V}(s_t, g) \big)
$$
$$
+ \big( ||\nabla_s V_{\boldsymbol{\theta}_V}(s_t, g)|| \cdot S(s_t) - 1 \big)^2 \Big]. \tag{14}
$$

The hierarchical policy extraction step follows Park et al. [10] and leverages, for both $\pi_{\boldsymbol{\theta}_{hi}}^{hi}$ and $\pi_{\boldsymbol{\theta}_{lo}}^{lo}$, an advantage-weighted regression-style objective:

$$
J_{\pi^{hi}}(\boldsymbol{\theta}_{hi}) = \mathbb{E}_{(s_t, s_{t+k}) \sim \mathcal{D}, g \sim \mathcal{P}_g} \Big[ \exp(\beta \cdot \tilde{A}^{hi}(s_t, s_{t+k}, g)) \log \pi_{\boldsymbol{\theta}_{hi}}^{hi}(s_{t+k}|s_t, g) \Big], \tag{15}
$$

$$
J_{\pi^{lo}}(\boldsymbol{\theta}_{lo}) = \mathbb{E}_{(s_t, a_t, s_{t+1}, s_{t+k}) \sim \mathcal{D}} \Big[ \exp(\beta \cdot \tilde{A}^{lo}(s_t, a_t, s_{t+k})) \log \pi_{\boldsymbol{\theta}_{lo}}^{lo}(a_t|s_t, s_{t+k}) \Big], \tag{16}
$$

where $\beta$ is an inverse temperature hyperparameter and $\tilde{A}^{hi}(s_t, s_{t+k}, g)$ and $\tilde{A}^{lo}(s_t, a_t, s_{t+k})$ are respectively approximated as $V_{\boldsymbol{\theta}_V}(s_{t+k}, g) - V_{\boldsymbol{\theta}_V}(s_t, g)$ and $V_{\boldsymbol{\theta}_V}(s_{t+1}, s_{t+k}) - V_{\boldsymbol{\theta}_V}(s_t, s_{t+k})$. The full pseudocode for Eik-HIQL is provided in Algorithm 1. Furthermore, a function written in JAX on how to compute the gradient $\nabla_s V_{\boldsymbol{\theta}_V}$ in (14) is summarized in Algorithm 2. Finally, Table 3 reports the hyperparameter values most commonly used in our experiments. For more implementation details, refer to our GitHub repository[2].

---

**Algorithm 1** Eikonal-regularized Hierarchical Implicit Q-Learning (Eik-HIQL)

---

**Input:** Offline dataset $\mathcal{D}$, value function $V_{\boldsymbol{\theta}_V}$, target value function $V_{\bar{\boldsymbol{\theta}}_V}$, high-level policy $\pi_{\boldsymbol{\theta}_{hi}}^{hi}$, low-level policy $\pi_{\boldsymbol{\theta}_{lo}}^{lo}$, speed profile $S$, expectile factor $\iota$, discount factor $\gamma$, inverse temperature parameter $\beta$, learning rates $\alpha_V$, $\alpha_{hi}$, $\alpha_{lo}$, target update rate $\tau$
**while** not converged **do**
    $(s_t, s_{t+1}, g) \sim \mathcal{D}$
    Update $V_{\boldsymbol{\theta}_V}$ minimizing $\mathcal{L}_V(\boldsymbol{\theta}_V)$ in (14) with learning rate $\alpha_V$
    $\bar{\boldsymbol{\theta}}_V \leftarrow (1 - \tau)\bar{\boldsymbol{\theta}}_V + \tau \boldsymbol{\theta}_V$
**end while**
**while** not converged **do**
    $(s_t, s_{t+k}, g) \sim \mathcal{D}$
    Update $\pi_{\boldsymbol{\theta}_{hi}}^{hi}$ maximizing $J_{\pi^{hi}}(\boldsymbol{\theta}_{hi})$ in (15) with learning rate $\alpha_{hi}$
**end while**
**while** not converged **do**
    $(s_t, a_t, s_{t+1}, s_{t+k}) \sim \mathcal{D}$
    Update $\pi_{\boldsymbol{\theta}_{lo}}^{lo}$ maximizing $J_{\pi^{lo}}(\boldsymbol{\theta}_{lo})$ in (16) with learning rate $\alpha_{lo}$
**end while**

---

**Algorithm 2** Compute $\nabla_s V_{\boldsymbol{\theta}_V}$

---

**Input:** states $s$, goals $g$, network parameters $\boldsymbol{\theta}_V$
**Define** FORWARD$(s, g, \boldsymbol{\theta}_V)$:
    **return** NETWORK.SELECT$(V_{\boldsymbol{\theta}_V})(s, g, \text{params} = \boldsymbol{\theta}_V)$
grad_s $\leftarrow$ JAX.VMAP(JAX.GRAD(FORWARD, argnums $= 0$), in_axes $= (0, 0, \textbf{None}))(s, g, \boldsymbol{\theta}_V)$
**return** grad_s

---

[2]https://github.com/VittorioGiammarino/Eik-HIQL

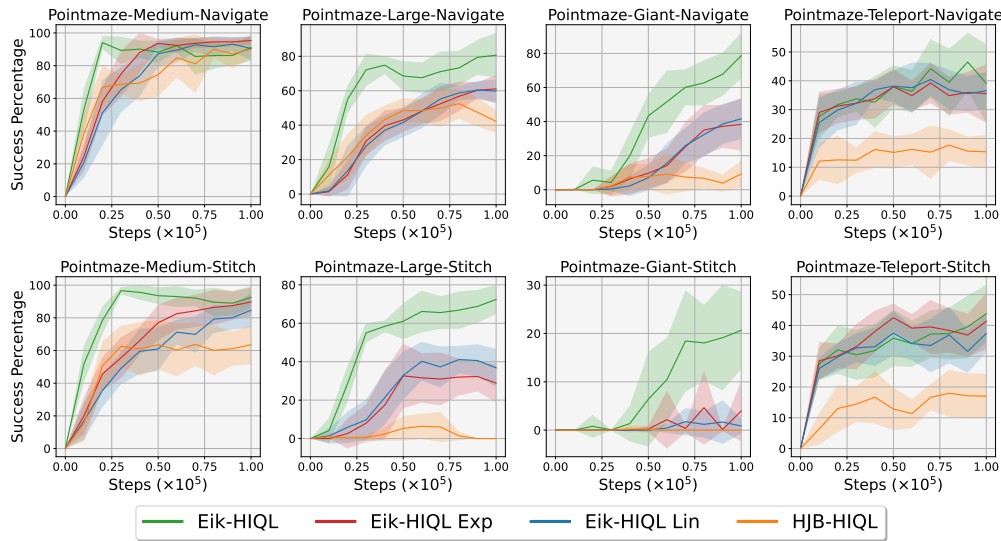

Figure 6: Learning curves for the speed profile ablation and the comparison with an HJB regularizer in Table 1. Plots show the average success percentage per evaluation across seeds as a function of training steps.

Table 3: Hyperparameter values for Eik-HIQL.

| Hyperparameter Name | Value |
|---|---|
| Decay rate ($\lambda$) | 1.0 |
| Minimum speed ($S_{\min}$) | 0.1 |
| Discount factor ($\gamma$) | 0.99 |
| Batch size ($B$) | 1024 |
| Optimizer | Adam |
| Learning rates $\alpha_V, \alpha_{hi}, \alpha_{lo}$ | $3 \cdot 10^{-4}$ |
| Target update rate ($\tau$) | 0.005 |
| Expectile factor ($\iota$) | 0.7 |
| Inverse temperature parameter ($\beta$) | 3.0 |

# E   Learning Curves

Fig. 6 shows the complete learning curves, plotted as a function of training steps, for the experiments reported in Table 1. Figs. 7, 8, 9, 10, and 11 display the learning curves for the `pointmaze`, `antmaze`, `humanoidmaze`, `antsoccer`, and `manipulation` experiments in Table 2, respectively.

All experiments were conducted on a single NVIDIA RTX 3090 GPU (24 GB VRAM), using a local server equipped with a 12th Gen Intel i7-12700F CPU, 32 GB RAM. No cloud services or compute clusters were used. Each individual experimental run required approximately 4 hours of compute time on the GPU.

# F   Additional Experiments

In the following, we present additional experiments demonstrating that our Eikonal regularizer can be seamlessly integrated with a broad range of temporal-difference (TD)-based GCRL algorithms. In particular, we apply it to Goal-Conditioned variants of IQL [23] and IVL [39, 40], yielding Eik-GCIQL and Eik-GCIVL, respectively. The corresponding results are summarized in Table 4, with learning curves provided in Fig. 12 and 13. These experiments confirm the same conclusions drawn in the main paper from the comparison between Eik-HIQL and HIQL (Table 2), and further support our claim that the Eikonal regularizer can be successfully combined with diverse TD-based algorithms.

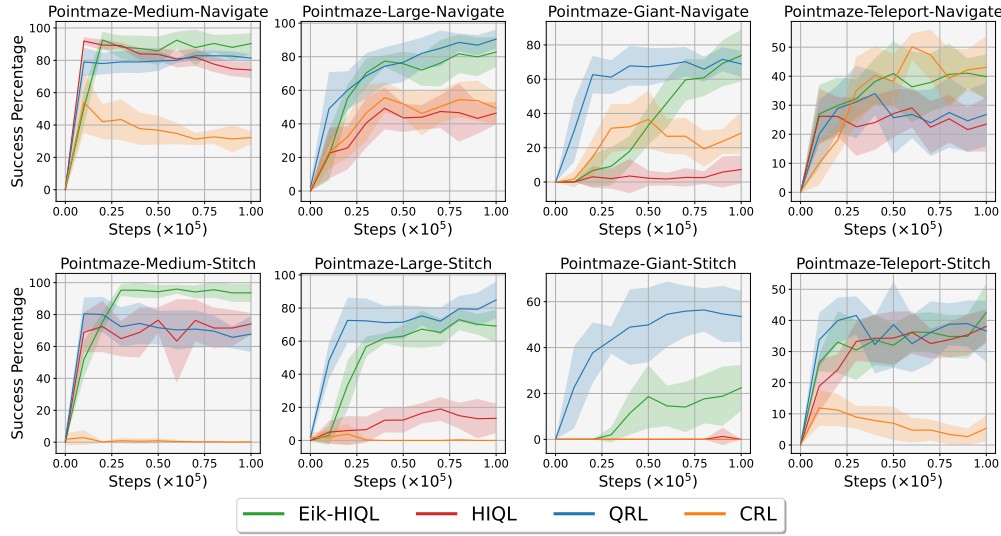

Figure 7: Learning curves for the `pointmaze` experiments in Table 2. Plots show the average success percentage per evaluation across seeds as a function of training steps.

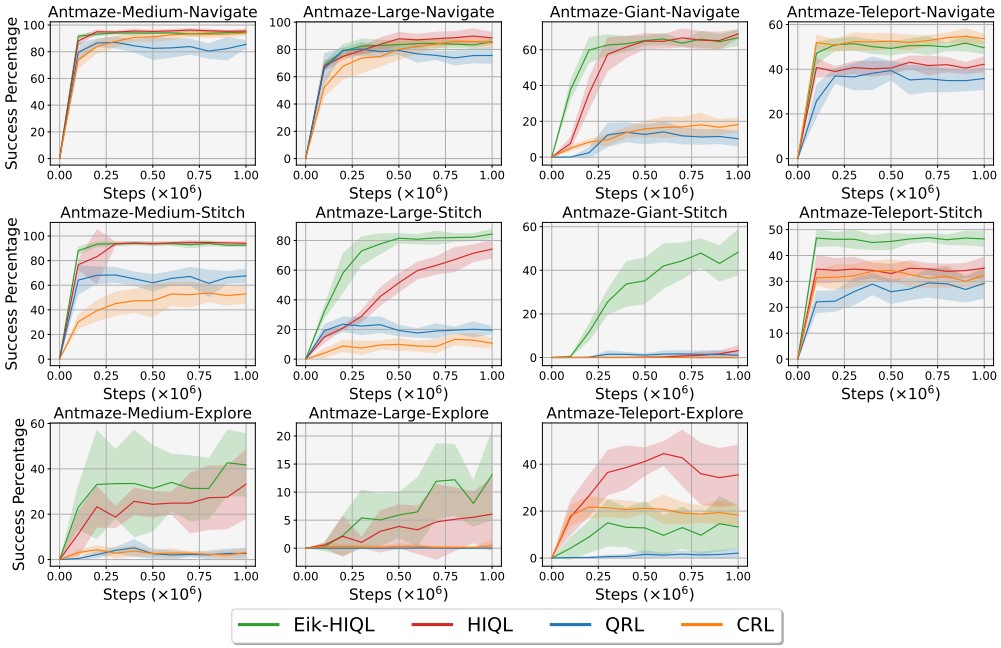

Figure 8: Learning curves for the `antmaze` experiments in Table 2. Plots show the average success percentage per evaluation across seeds as a function of training steps.

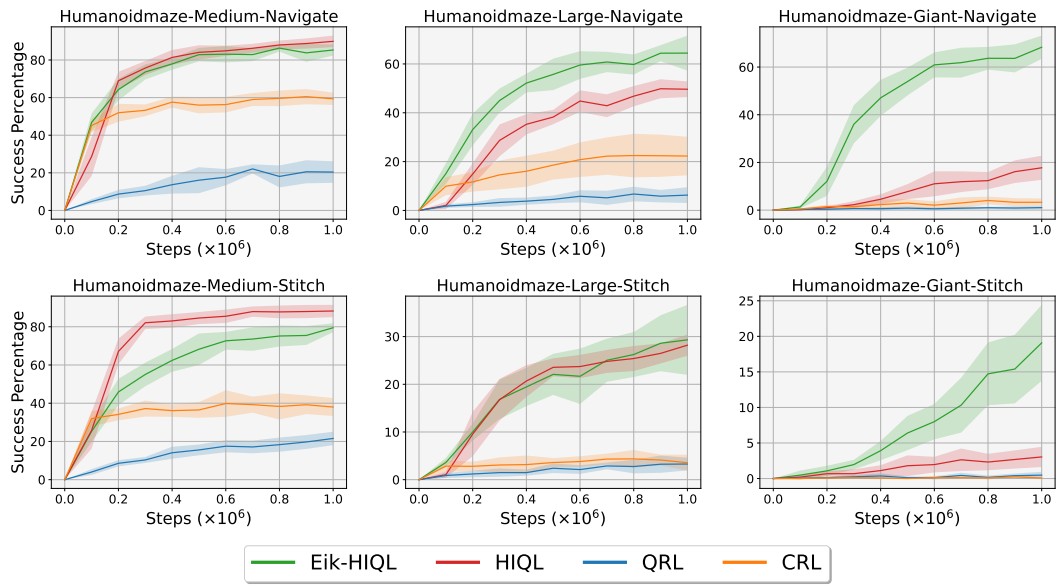

Figure 9: Learning curves for the `humanoidmaze` experiments in Table 2. Plots show the average success percentage per evaluation across seeds as a function of training steps.

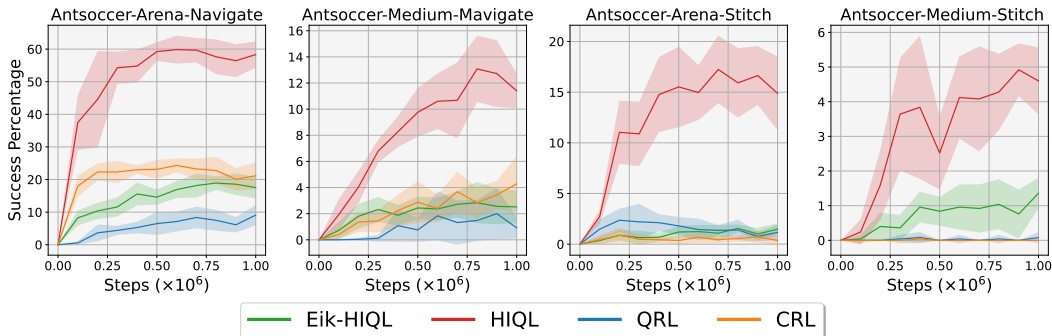

Figure 10: Learning curves for the `antsoccer` experiments in Table 2. Plots show the average success percentage per evaluation across seeds as a function of training steps.

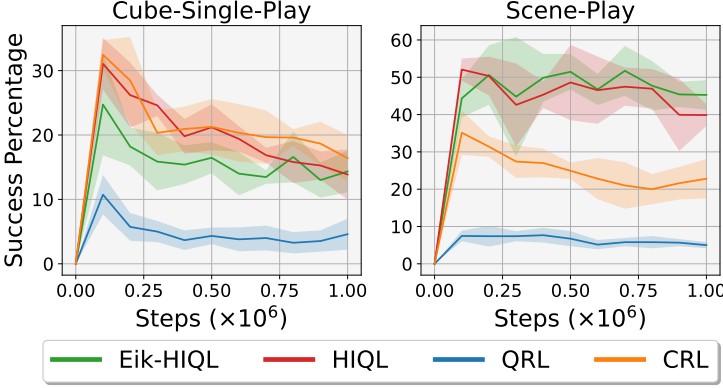

Figure 11: Learning curves for the `manipulation` experiments in Table 2. Plots show the average success percentage per evaluation across seeds as a function of training steps.

Table 4: Summary of the experiments with different TD-based GCRL algorithms. All agents are trained for **100,000 training steps** using 10 seeds. We report the mean and standard deviation across seeds for the best evaluation achieved during training. For each seed, evaluations are conducted over 5 different random goals, as designed in Park et al. [11], with the learned policy tested for 50 episodes per goal. Results within $95\%$ of the best value are written in **bold**.

| Environment | Dataset Type | Maze Dimension | GCIQL | Eik-GCIQL | GCIVL | Eik-GCIVL |
|---|---|---|---|---|---|---|
| pointmaze | navigate | medium | $60 \pm 1$ | $59 \pm 9$ | $63 \pm 6$ | $\mathbf{90 \pm 5}$ |
| | | large | $39 \pm 1$ | $60 \pm 9$ | $38 \pm 5$ | $\mathbf{82 \pm 39}$ |
| | | giant | $0 \pm 0$ | $2 \pm 4$ | $0 \pm 0$ | $\mathbf{86 \pm 11}$ |
| | | teleport | $29 \pm 5$ | $25 \pm 12$ | $38 \pm 5$ | $\mathbf{49 \pm 4}$ |
| | stitch | medium | $41 \pm 11$ | $56 \pm 6$ | $57 \pm 9$ | $\mathbf{95 \pm 4}$ |
| | | large | $25 \pm 8$ | $22 \pm 3$ | $11 \pm 8$ | $\mathbf{67 \pm 9}$ |
| | | giant | $0 \pm 0$ | $0 \pm 0$ | $0 \pm 0$ | $\mathbf{23 \pm 10}$ |
| | | teleport | $28 \pm 5$ | $25 \pm 3$ | $\mathbf{41 \pm 5}$ | $38 \pm 3$ |
| antmaze | navigate | medium | $27 \pm 4$ | $25 \pm 6$ | $36 \pm 5$ | $\mathbf{50 \pm 5}$ |
| | | large | $9 \pm 3$ | $7 \pm 2$ | $\mathbf{16 \pm 4}$ | $\mathbf{15 \pm 3}$ |
| | | giant | $0 \pm 0$ | $0 \pm 0$ | $0 \pm 0$ | $0 \pm 0$ |
| | | teleport | $24 \pm 3$ | $23 \pm 2$ | $\mathbf{32 \pm 5}$ | $30 \pm 3$ |
| | stitch | medium | $19 \pm 4$ | $21 \pm 5$ | $\mathbf{25 \pm 4}$ | $\mathbf{27 \pm 6}$ |
| | | large | $6 \pm 3$ | $3 \pm 3$ | $\mathbf{12 \pm 3}$ | $7 \pm 2$ |
| | | giant | $0 \pm 0$ | $0 \pm 0$ | $0 \pm 0$ | $0 \pm 0$ |
| | | teleport | $18 \pm 5$ | $23 \pm 3$ | $\mathbf{30 \pm 3}$ | $28 \pm 3$ |

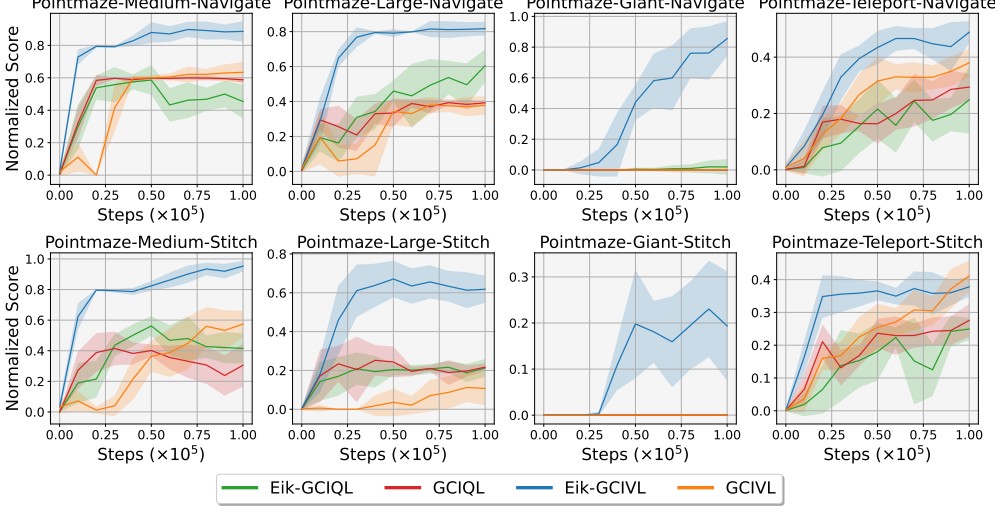

Figure 12: Learning curves for the pointmaze experiments in Table 4. Plots show the average success percentage per evaluation across seeds as a function of training steps.

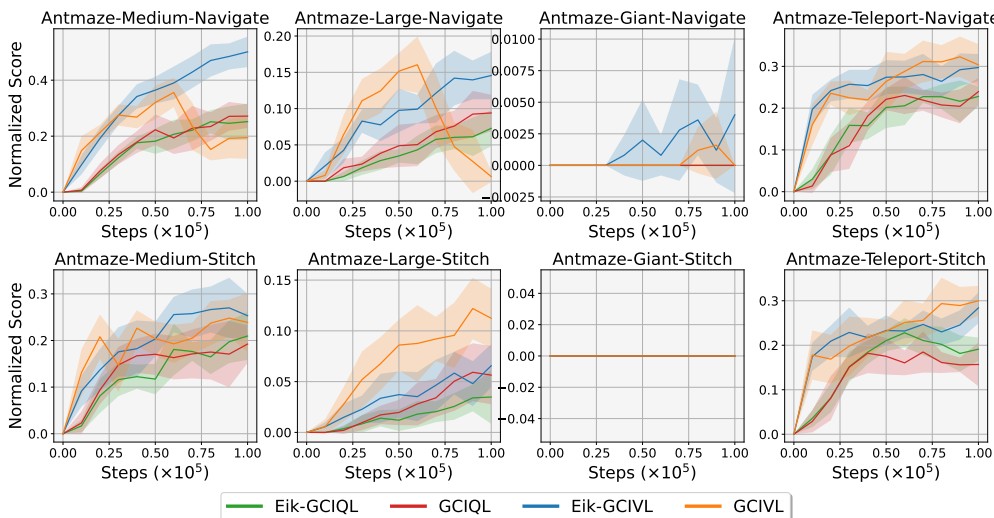

Figure 13: Learning curves for the `antmaze` experiments in Table 2. Plots show the average success percentage per evaluation across seeds as a function of training steps.

