# OpenReview forum: "Physics-informed Value Learner for Offline Goal-Conditioned Reinforcement Learning"
_NeurIPS.cc/2025/Conference — NeurIPS 2025 poster_

### Official Review · Reviewer_zhNd · 2025-06-25

**Clarity:** 3
**Significance:** 3
**Originality:** 2
**Rating:** 5
**Confidence:** 4

**Summary:**

Value estimation is a hard problem that can directly affect the performance and generalization of policies learned via offline RL, especially in the goal-conditioned (GC) setting. Prior work has explored using a physics-informed (PI) regularizer, via the Hamilton-Jacobi-Bellman (HJB) equation, which requires explicit access to system dynamics. Instead, this paper proposes the use of the Eikonal PDE, which is model free and induces a distance field structure in the estimated value function. The Eikonal PDE relates a speed profile function to a travel time function between a state and goal, via its spatial gradient. A high speed profile results in low travel time, and vice versa. The authors justify the use of the Eikonal PDE by relating it to the HJB equation. Specifically, they show that the corresponding Hamiltonian in the HJB equation is upper-bounded by an Eikonal residual, with the travel time function set to the value function and making a specific choice of local speed profile involving the dynamics and cost functions. Using this connection, they propose a general regularization term inspired by the Eikonal residual. It leaves the speed profile as a general function which can be chosen by the practitioner.

They implement this regularized value function estimation protocol in the context of hierarchical implicit Q-learning (HIQL), an algorithm for offline GCRL, and evaluate the approach on environments from OGbench, which contains multiple goal-conditioned tasks. They consider three variants of their algorithm, making different choices for the speed profile term, including a constant, exponential, and linear function of distance to obstacles and goals. Additionally, they compare to a regularization term inspired by the HJB equation, as used in prior work. On a specific pointmaze environment, across two types of datasets and four maze sizes, they find their approach, Pi-HIQL with a constant speed profile, outperforms the other variants and the HJB baseline. They then compare Pi-HIQL with a constant speed profile to standard HIQL across multiple environments. We see Pi-HIQL making substantial gains across multiple environments, especially in the case where it must stitch together trajectories from the dataset. However, they find in contact-rich domains that their approach underperforms standard HIQL. Finally, they compare to other offline GCRL algorithms and show that their method does better in long-horizon and large-scale environments.

**Questions:**

- Is the choice of a constant S(s) = 1 motivated by the goal-conditioned reward discussed in Section 3, which has a maximum norm of 1? Do you think the more complex alternatives would outperform the constant speed profile with more complex reward functions?
- What are some strategies for overcoming the limitations in environments with discontinuities? Would this primarily occur in the choice of distance function used in the speed profile?

**Ethical Concerns:**

["NO or VERY MINOR ethics concerns only"]

**Limitations:**

- The authors clearly state that this regularization is especially well-suited to environments without sharp discontinuities. They also show this in their evaluations, with their approach underperforming in contact-rich tasks.
- They also fully acknowledge that this regularization term requires access to obstacle locations, which may be a strong assumption in some settings.

**Quality:**

3

**Strengths And Weaknesses:**

Strengths:
- Offline RL and goal-conditioned RL are both important frameworks where improved value function estimation can have a big impact, and this is well-motivated in the introduction of the paper.
- The evaluations clearly show that their method outperforms both the unregularized version of HIQL and an HJB-inspired regularization term on many environments. They also consider three variants of their approach which make different choices for the speed profile function. And they compare to other choices of offline GCRL algorithms as well, making their evaluations fairly rigorous.
- In cases where their method underperforms, it is well justified by the assumptions made with their regularization strategy. The authors are upfront about these limitations, showing clearly when this approach is most applicable and when it may not be as useful.
- The connection to the HJB equation is well-formulated, and the evaluation does a good job comparing to an HJB-based regularization strategy.
- Figure 1 provides a nice visualization of how the physics-informed regularizer can improve value function estimation in the goal-conditioned setting.

Weaknesses:
- Modifying the choice of regularization term is a fairly minor, straight-forward change, especially in the context of prior work that regularize with the HJB equation. However, it is well-motivated in the paper and evaluated well.
- In section 5, the different choices of speed profiles make use of a distance function, d(s), which is never clearly defined. It is only visualized for the pointmaze environment in Figure 4. It would be helpful to have these terms clearly defined somewhere, preferably in an environment-specific way.
- While Figure 5 provides a nice visualization of the learned value function contours, I find it hard to see how "Pi_HIQL Exp and Pi-HIQL Lin display artifcats even near the goal". Some annotations of the plots to help illustrate this would be helpful. The main thing I can see is how Pi-HIQL HJB fails to capture maze geometry, which is crucial too.
- A minor point: I would suggest expanding acronyms in the introduction of the paper, even if they were defined in the abstract for clarity.

---

> ### Author Rebuttal · Authors · 2025-07-30
>
> Thank you very much for your detailed summary and thoughtful review. We truly appreciate your evaluation and the time you dedicated to providing constructive feedback.
>
> We have carefully addressed your comments and questions in the rebuttal, and we are grateful for your suggestions, which have helped improve the clarity and overall quality of our work.
>
> ## W2, W3, W4
> > In section 5, the different choices of speed profiles make use of a distance function, d(s), which is never clearly defined.
>
> > While Figure 5 provides a nice visualization of the learned value function contours, I find it hard to see how "Pi_HIQL Exp and Pi-HIQL Lin display artifcats even near the goal". Some annotations of the plots to help illustrate this would be helpful.
>
> > I would suggest expanding acronyms in the introduction of the paper, even if they were defined in the abstract for clarity.
>
>
> Thank you for pointing out these issues. We will revise the final version of the paper accordingly to address all the raised concerns.
>
> Specifically, we will explicitly define the distance function $d(s)$ in Appendix B, add annotations to Fig. 5 to highlight the observed artifacts in Pi_HIQL Exp and Pi-HIQL Lin, and expand all acronyms upon first use in the introduction to improve clarity for the reader.
>
> ## Q1
> > Is the choice of a constant S(s) = 1 motivated by the goal-conditioned reward discussed in Section 3, which has a maximum norm of 1? Do you think the more complex alternatives would outperform the constant speed profile with more complex reward functions?
>
> Thank you for this insightful question. We do believe that more expressive, state-dependent speed profiles $S(s)$ have the potential to outperform the constant choice $S(s)=1$ in settings with more complex requirements and reward functions. This is primarily because $S(s)$ enables direct shaping of the value function, introducing useful inductive biases into the structure of $V(s,g)$. Such biases can lead to a more efficient approximation of the optimal value function $V^*(s,g)$.
>
> While the constant profile performs well in our current setting, we are actively working on extending our framework to richer environments, where the advantages of structured speed functions can be more clearly observed. We look forward to sharing results in this direction in future work.
>
> ## Q2
> > What are some strategies for overcoming the limitations in environments with discontinuities? Would this primarily occur in the choice of distance function used in the speed profile?
>
> Thank you for asking this important question. We agree that the choice of the speed profile can help mitigate some of the challenges posed by discontinuities in the value function, although it may not provide a general solution to the problem.
>
> In environments where the optimal value function $V^*(s, g)$ exhibits discontinuities and is not globally differentiable, our Eikonal regularizer, which relies on gradient information, should ideally be applied only on differentiable regions of the state space. One possible strategy is to design a speed profile that acts as a switching mechanism, effectively disabling the regularization term in regions where discontinuities are likely to occur.
>
> An alternative direction involves using representation learning to map the state into a space where the value function is smoother or globally differentiable. We consider this to be a promising avenue for future work and are currently exploring this direction in ongoing research.

---

### Official Review · Reviewer_3kEX · 2025-06-28

**Clarity:** 2
**Significance:** 2
**Originality:** 2
**Rating:** 4
**Confidence:** 3

**Summary:**

This paper presents **Pi-HIQL**, a physics-informed regularization technique for offline goal-conditioned reinforcement learning. The method introduces a geometric inductive bias derived from the Eikonal partial differential equation (PDE), aiming to improve value learning and generalization in long-horizon and sparse-reward tasks.

**Questions:**

1. In Equation 5, why must the expression equal zero? Could delta_t be zero in some cases to satisfy the equation trivially, and if so, what would that imply for learning?
2. In the navigation task, the robot must handle contact dynamics with the ground, which can induce discontinuities in state transitions. Why is the proposed regularizer not affected by such discontinuities?

**Ethical Concerns:**

["NO or VERY MINOR ethics concerns only"]

**Final Justification:**

The rebuttal solved my concerns about the theoretical parts.

**Limitations:**

Yes

**Quality:**

2

**Strengths And Weaknesses:**

**Strengths:**

- The paper offers a compelling intuition and theoretical foundation through the use of an Eikonal-based regularizer.

**Weaknesses:**

- Certain aspects of the method require further clarification. For example, the proof and derivation of Proposition 4.1 are incomplete; specifically, the simplification to the HJB equation is omitted. The proposition's correctness is questionable, as minimizing a sum does not necessarily yield a result smaller than minimizing the terms independently.
- Some critical design choices are left unmotivated — in particular, the selection of the speed function S(s)S(s)S(s). Using a constant speed seems oversimplified, and it's unclear how this choice affects the regularization dynamics.
- In practice, the proposed method reduces to regularizing the norm of the value function's gradient. While the Eikonal perspective provides one interpretation, alternative formulations exist, and the benefits of adopting this particular regularizer over others are not fully justified.

---

> ### Author Rebuttal · Authors · 2025-07-30
>
> Thank you for your detailed review. We appreciate your consideration of our submission and the feedback you provided.
>
> We have taken your comments seriously and responded to each of your concerns in detail. If our explanations address your doubts and clarify the issues you raised, we kindly ask you to consider revisiting your score.
>
> Please do not hesitate to reach out if there are any remaining uncertainties or further questions. Thank you again for your time and for giving us the opportunity to improve our work.
>
> ## W1, Q1
> > The proof and derivation of Proposition 4.1 are incomplete; specifically, the simplification to the HJB equation is omitted.
>
> > In Equation 5, why must the expression equal zero? Could delta_t be zero in some cases to satisfy the equation trivially, and if so, what would that imply for learning?
>
> Thank you for pointing this out. We will revise the final version of the paper to include additional steps leading to the HJB equation in Eq. (5). Briefly, by substituting the Taylor expansion $V(s(t+\Delta t), g) = V(s, g) + \nabla_s V(s,g)^{\intercal}f(s,a)\Delta t + O(\Delta t^2)$ into the principle of optimality in Eq.(4), we obtain:
>
> $$V(s,g) = \inf_{a \in \mathcal{A}}[c(s,a)\Delta t + V(s,g) + \nabla_s V(s,g)^{\intercal}f(s,a)\Delta t  + O(\Delta t^2)].$$
>
> Subtracting $V(s,g)$ on both sides and dividing by $\Delta t$ gives:
>
> $$0 = \inf_{a \in \mathcal{A}}[c(s,a) + \nabla_s V(s,g)^{\intercal}f(s,a) + O(\Delta t)].$$
>
> Taking the limit as $\Delta t \to 0$, we recover Eq. (5):
>
> $$ \inf_{a \in \mathcal{A}} [c(s,a) + \nabla_s V(s,a)^{\intercal}f(s,a)]=0.$$
>
> Regarding the second question: $\Delta t$ serves as a small positive quantity in a first-order approximation, and the derivation considers the limiting behavior as $\Delta t \to 0$. This is standard in continuous-time control theory.
>
> >The proposition's correctness is questionable, as minimizing a sum does not necessarily yield a result smaller than minimizing the terms independently.
>
> We believe there may have been a misunderstanding. As outlined in the proof, we first apply the Cauchy-Schwarz inequality to the argument of the minimization:
>
> $$ c(s,a) + \nabla_s V(s,g)^{\intercal}f(s,a) \leq c(s,a) + ||\nabla_s V(s,g)||||f(s,a)||.$$
>
> Then, by defining
>
> $$F^*(s)=\sup_{a \in \mathcal{A}}||f(s,a)||,$$
>
> we can further upper bound the right-hand-side and obtain:
>
> $$ c(s,a) + \nabla_s V(s,g)^{\intercal}f(s,a) \leq c(s,a) + ||\nabla_s V(s,g)||F^*(s).$$
>
> Finally, applying the infimum over $a \in \mathcal{A}$ on both sides yields:
>
> $$\inf_{a \in \mathcal{A}}[c(s,a) + \nabla_s V(s,g)^{\intercal}f(s,a)] \leq \inf_{a \in \mathcal{A}} [c(s,a) + ||V(s,g)||F^*(s)],$$
>
> where the result in Eq. (6) is obtained by defining $c^*(s) = \inf_{a \in \mathcal{A}} c(s,a)$.
>
> We hope this clarifies the derivation. Please also refer to [1] for a study that highlights the Eikonal PDE as a special case of HJB PDE . To avoid any confusion, we will include the full step-by-step derivation in the final version of the paper.
>
> ## W2
> > The selection of the speed function $S(s)$. Using a constant speed seems oversimplified, and it's unclear how this choice affects the regularization dynamics.
>
> We respectfully disagree with this assessment. The choice of a constant speed function has been carefully motivated throughout the paper. In particular, lines 225-235 and 275-288 discuss the rationale behind this design choice, emphasizing its simplicity and effectiveness.
>
> Moreover, Table 1 provides empirical evidence that supports its use, showing consistent improvements when the regularizer is applied.
>
> Finally, Fig. 1 illustrates the impact of the regularizer on the learned value function, demonstrating that even with a constant $S(s)$, the regularization yields meaningful structural benefits.
>
> ## W3
> > In practice, the proposed method reduces to regularizing the norm of the value function's gradient. While the Eikonal perspective provides one interpretation, alternative formulations exist, and the benefits of adopting this particular regularizer over others are not fully justified.
>
> To the best of our knowledge, no prior work has proposed regularizers specifically designed for value function learning in the Goal-Conditioned RL setting, particularly those grounded in physics-informed principles such as the Eikonal formulation.
>
> We have reviewed and summarized the most relevant literature in our Related Work section. If there are additional approaches we may have overlooked, we would greatly appreciate any specific references and will be glad to acknowledge and discuss them in the final version of the paper.
>
> ## Q2
> > In the navigation task, the robot must handle contact dynamics with the ground, which can induce discontinuities in state transitions. Why is the proposed regularizer not affected by such discontinuities?
>
> Thank you for asking this question. We are actively studying this phenomenon and hope to provide a more comprehensive answer in future extensions of our work.
>
> To the best of our understanding, in tasks such as AntSoccer and robotic manipulation, the presence of external objects induces more severe and non-local discontinuities in the state space. These arise from contact events, such as collisions and object interactions, and are directly encoded in the state representation. As a result, the value function exhibits more pronounced discontinuities compared to standard locomotion tasks.
>
> In contrast, in locomotion tasks, although contact dynamics are present, the transitions tend to be internally consistent and more predictable. This leads to smoother value functions, which align better with the assumptions of our regularizer. These observations explain its stronger performance in locomotion settings and highlight the need for more structured representations or multi-body modeling when applying similar techniques to manipulation-like domains.
>
> **References**
>
> [1] Cacace S, Cristiani E, Falcone M. Can Local Single-Pass Methods Solve Any Stationary Hamilton--Jacobi--Bellman Equation?. SIAM Journal on Scientific Computing. 2014;36(2):A570-87.

---

> > ### Comment · Reviewer_3kEX · 2025-08-05
> >
> > Thank the authors for the rebuttal, which has clearly answered my questions. I will raise my score.

---

### Official Review · Reviewer_XkBr · 2025-06-30

**Clarity:** 3
**Significance:** 3
**Originality:** 3
**Rating:** 5
**Confidence:** 4

**Summary:**

This paper introduces Physics-Informed HIQL (PI-HIQL), an offline goal-conditioned reinforcement learning (GCRL) approach that integrates a physics-informed (PI) regularizer derived from the Eikonal partial differential equation (PDE). The authors emphasize that existing offline GCRL methods struggle with value function estimation due to limited data coverage, particularly in sparse environments. To address this, PI-HIQL augments Hierarchical Implicit Q-Learning (HIQL) with an Eikonal regularizer that enforces a distance-like structure on the goal-conditioned value function (GCVF). The experiment results demonstrate the effectiveness of the proposed approach.

**Questions:**

1. Have you examined the sample efficiency of the PI-HIQL? Since including PI during learning could essentially include more informative information during training the policy. Note that I'm not asking for additional results, just for discussion purpose.

**Ethical Concerns:**

["NO or VERY MINOR ethics concerns only"]

**Final Justification:**

The paper is well-structured and effectively presents the main idea. Incorporating physical principles into offline RL is a conceptually novel and promising direction. Prior work has shown that physics-informed biases can enhance representation learning and regularization in decision-making, and this work extends this paradigm to value function estimation in GCRL. There are no explicit limitations overall, thus I keep the score as 5

**Limitations:**

Yes, the paper includes thorough discussions of the limitations.

**Paper Formatting Concerns:**

There are no major formatting issues in this paper.

**Quality:**

3

**Strengths And Weaknesses:**

**Pros:**
1. The paper is well-structured and effectively presents the main idea.

2. Incorporating physical principles (via the Eikonal PDE) into offline RL is a conceptually novel and promising direction. Prior work has shown that physics-informed biases can enhance representation learning and regularization in decision-making, and this work extends this paradigm to value function estimation in GCRL.

3. The method demonstrates that a simple, physics-inspired regularization term, when added to an SOTA GCRL algorithm (HIQL), substantially improves performance across diverse navigation tasks. This underscores the regularizer’s utility as a lightweight yet impactful modification for offline RL frameworks.

**Cons:**

From my perspective, there are no significant drawbacks. But one minor clarification problem in the content:
* In line 73, the phrase "in model-based or Koopman inspired frameworks" could be more precise. It is suggested to explicitly state how these frameworks leverage underlying dynamical properties: ​​"utilizing the Koopman operator framework and incorporating Time-reversal symmetry information of the dynamics"​​ would enhance technical clarity.

---

> ### Author Rebuttal · Authors · 2025-07-30
>
> Thank you very much for your review. We truly appreciate the positive evaluation and the time you took to provide constructive feedback.
>
> We have carefully addressed your comments and questions in the rebuttal, and we are grateful for your suggestions, which helped us improve the clarity and quality of our work.
>
> ## W1
> > In line 73, the phrase "in model-based or Koopman inspired frameworks" could be more precise.
>
> Thank you for pointing this out. We agree with the reviewer’s suggestion and will revise the phrasing in line 73 to improve technical clarity.
>
> ## Q1
> > Have you examined the sample efficiency of the PI-HIQL?
>
> Thank you for raising this important point. While we did not include a dedicated sample efficiency analysis, we refer the reviewer to the learning curves in Appendix D, and in particular to the results on the AntMaze benchmark in Fig. 8, where both Pi-HIQL and HIQL eventually succeed. In these plots, the impact of the regularizer on sample efficiency can be better appreciated.
>
> For example, in tasks such as *antmaze-giant-navigate* and *antmaze-large-stitch*, we observe that Pi-HIQL converges significantly earlier than HIQL, indicating improved sample efficiency. In other tasks, such as *antmaze-giant-stitch* and in the humanoid setting, we observe that the Pi regularizer not only improves efficiency but also enables learning altogether, where HIQL alone fails to make progress.

---

### Official Review · Reviewer_zAXA · 2025-07-02

**Clarity:** 4
**Significance:** 3
**Originality:** 3
**Rating:** 5
**Confidence:** 3

**Summary:**

The papers proposes a new smoothness-inducing regularizer designed for Offline GCRL based on the Eikonal PDE. It implements HIQL extended with the Eikonal regularizer and demonstrates that it achieves SoTA on some robotic benchmarks.

**Questions:**

1. Application of Eikonal regularizer to all GCRL: What about GCRL that is not hierarchical? How can this regularizer be applied there?
2. It would be interesting see an example - even artificially made up - in which S(s) other than 1 induces useful structural bias and helps in some way with the training.
3. I suppose the weird behaviour of HIQL on the Figure 1 is because it is not fully trained yet and the value function resembles expected, optimal value function at the end. How can authors guarantee that the effect observed in Figure 1 is not a consequence of suboptimal hyperparameters?
4. Training curves would be interesting to see to understand the Figure 1 in more details.
5. It would also be interesting to consider the performance of the eikonal regularizer from the perspective of sample efficiency.
6. For the non-Lipschitz environments mentioned in the limitations. Does Eikonal regularizer improve the performance at the beginning of the training?

**Ethical Concerns:**

["NO or VERY MINOR ethics concerns only"]

**Final Justification:**

My initial concerns centered around the generality and applicability of the regularizer, its evaluation scope, and the interpretation of results. The authors have addressed these in a thorough and convincing manner:

On generality: The authors clarified that the regularizer is not limited to hierarchical methods. They provided new empirical results demonstrating its applicability to non-hierarchical TD-based methods (GCIQL and GCIVL), confirming its broader compatibility across the GCRL landscape. This directly resolves my concern (W1, W2, Q1).

On benchmark diversity: While I initially noted the evaluation as limited (W3), the authors justifiably argue that OGBench is a comprehensive benchmark spanning various environments and challenges. The added experiments further strengthen this claim.

The authors also acknowledge the limitations in non-Lipschitz environments and outline this as an avenue for future work.

Taken together, these clarifications and additional experiments improve the strength and clarity of the paper. The work is methodologically sound, well-written, and demonstrates clear potential for impact in offline RL, especially in the under-explored space of structure-inducing regularizers.

**Limitations:**

yes

**Quality:**

3

**Strengths And Weaknesses:**

## Strengths

1. Paper is well written with properly defined background terminology.
2. The method created with Eikonal regularizer achieves SoTA on the inspected tasks.
3. The method nicely utilizes hierarchical nature of high-level policy to induce a structural bias with the regularizer that uses subgoals.
4. The limitations are discussed explicitly.

## Weaknesses

> The proposed regularizer is broadly compatible with temporal-difference-based value learning and can be integrated into existing Offline GCRL algorithms.
>
1. The regularizer is evaluated only when combined with the HIQL, for such a general claims it should be evaluated on other GCRL methods. For example table 2 could also have Pi-QRL, Pi-CRL. Even if they don’t reach SoTA, it is an important information whether they improve their baselines.
2. I’m not sure if the proposed regularizer can even be applied to any GCRL method. It requires an action to be a state (a subgoal), hence requires high-level policy and therefore hierarchical reinforcement learning. (See question no.1)
3. The evaluation is quite limited in terms of benchmarks
4. The benefit of the regularizer seem to dissipate in more complex environments, which poses risks to scalability and applicability of the method in real-life scenarios.
5. I think “Physics-informed” is a very general name that does not sound very specific to the regularizer. For example HJB regularized HIQL could also be called physics-informed. Maybe put “Eikonal” in the name?

---

> ### Author Rebuttal · Authors · 2025-07-30
>
> Thank you for reviewing our work. We appreciate your constructive feedback and the time you dedicated to evaluating our submission.
>
> We have carefully addressed each of your comments and clarified the points you raised in our rebuttal. If our responses have satisfactorily resolved your concerns, we would be grateful if you could consider raising your score.
>
> If there are still aspects that remain unclear, please feel free to reach out, we are happy to provide further clarifications. Thank you again for your review and for helping us improve our work.
>
> ## W1, W2, Q1
>
> > The regularizer is evaluated only when combined with the HIQL, for such a general claim it should be evaluated on other GCRL methods.
>
> > I’m not sure if the proposed regularizer can even be applied to any GCRL method. It requires an action to be a state (a subgoal), hence requires high-level policy and therefore hierarchical reinforcement learning. (See question no.1)
>
> Thank you for raising this important point. We would like to clarify that the proposed Eikonal regularizer, as defined in Eq. (9), is compatible with any Temporal Difference (TD)-based Goal-Conditioned value learning method, such as the formulation shown in Eq. (8).
>
> Importantly, it does not require a hierarchical actor or subgoal-based high-level policy, and is therefore applicable beyond just HIQL. In our work, we apply it to HIQL because it represents a strong baseline and serves as an ideal candidate for demonstrating the benefits of our approach ( as mentioned in lines 48–51).
>
> To reinforce this point, we have conducted additional experiments using a Goal-Conditioned adaptation of two non-hierarchical algorithms:
>
> 1. Goal-Conditioned Implicit Q-Learning (GCIQL) [1]
> 2. Goal-Conditioned Implicit V-Learning (GCIVL) [2,3]
>
> In both cases, we applied our Eikonal regularizer (Pi-GCIQL and Pi-GCIVL, respectively), without any hierarchical structure or high-level policy. For these experiments we follow the same procedure described in the caption of Table 1. We will include the following results in the final version of the paper.
>
> | Environment | GCIQL | Pi-GCIQL | GCIVL | Pi-GCIVL |
> | ------------------ | --------- | ------------ | --------- | ------------- |
> | *pointmaze-medium-navigate* | 60 +- 1 | 59 +- 9 | 63 +- 6 | **90 +- 5** |
> | *pointmaze-large-navigate* | 39 +- 1 | 60 +- 9 | 38 +- 5 | **82 +- 39** |
> | *pointmaze-giant-navigate* | 0 +- 0 | 2 +- 4 | 0 +- 0 | **86 +- 11** |
> | *pointmaze-teleport-navigate* | 29 +- 5 | 25 +- 12 | 38 +- 5 | **49 +- 4** |
> | *pointmaze-medium-stitch* | 41 +- 11 | 56 +- 6 | 57 +- 9 | **95 +- 4** |
> | *pointmaze-large-stitch* | 25 +- 8 | 22 +- 3 | 11 +- 8 | **67 +- 9** |
> | *pointmaze-giant-stitch* | 0 +- 0 | 0 +- 0 | 0 +- 0 | **23 +- 10** |
> | *pointmaze-teleport-stitch* | 28 +- 5 | 25 +- 3 | **41 +- 5** | **38 +- 3** |
> | *antmaze-medium-navigate* | 27 +- 4 | 25 +- 6 | 36 +- 5 | **50 +- 5** |
> | *antmaze-large-navigate* | 9 +- 3 | 7 +- 2 | **16 +- 4** | **15 +- 3** |
> | *antmaze-giant-navigate* | 0 +- 0 | 0 +- 0 | 0 +- 0 | 0 +- 0 |
> | *antmaze-teleport-navigate* | 24 +- 3 | 23 +- 2 | **32 +- 5** | **30 +- 3** |
> | *antmaze-medium-stitch* | 19 +- 4 | 21 +- 5 | **25 +- 4** | **27 +- 6** |
> | *antmaze-large-stitch* | 6 +- 3 | 3 +- 3 | **12 +- 3** | 7 +- 2 |
> | *antmaze-giant-stitch* | 0 +- 0 | 0 +- 0 | 0 +- 0 | 0 +- 0 |
> | *antmaze-teleport-stitch* | 18 +- 5 | 23 +- 3 | **30 +- 3** | **28 +- 3** |
>
> Furthermore, we have also worked on Physics-informed extensions of QRL and CRL. However, not being these two algorithms purely based on TD learning, more considerations are required and we defer this to future work.
>
> ## W3
> > The evaluation is quite limited in terms of benchmarks
>
> We would like to emphasize that the OGBench benchmark [4] is a recently introduced and comprehensive suite that spans a diverse set of agents, environments, and offline datasets.
>
> While we acknowledge the importance of broad evaluation, we believe that incorporating additional benchmarks would not have substantially enhanced the insights provided. The chosen tasks already allowed us to highlight both the strengths and limitations of our approach across varied scenarios, offering a meaningful and representative assessment.
>
> ## W4
> > The benefit of the regularizer seem to dissipate in more complex environments
>
> We respectfully disagree with this observation. In our experiments, environment complexity can be understood along three key axes: (1) agent complexity, (2) maze size, and (3) dataset type.
>
> As shown in Table 2, our proposed Pi-HIQL consistently outperforms HIQL across all three dimensions. For instance, in high-complexity scenarios such as *humanoid-giant-navigate* and *antmaze-giant-stitch*, Pi-HIQL demonstrates clear advantages over HIQL. These results highlight that the benefits of our regularizer not only persist but become more pronounced as the tasks grow more challenging.
>
> ## W5
> > “Physics-informed” is a very general name that does not sound very specific to the regularizer.
>
> We appreciate the reviewer’s feedback and understand the concern about the generality of the term. While “Physics-informed” is indeed broad, we chose it to reflect the fact that our regularizer is grounded in principles from physics-based optimal control, specifically through PDE-inspired constraints.
>
> That said, we agree it is important to avoid confusion, and we will clarify the terminology in the final version.
>
> ## Q2
> > It would be interesting see an example - even artificially made up - in which S(s) other than 1 induces useful structural bias and helps in some way with the training.
>
> We agree this is a very relevant point. Unfortunately, in our current experimental setup, safety violations such as collisions with obstacles are not directly penalized. We conjecture that this lack of penalty is the primary reason why the structural bias introduced by varying $S(s)$ does not yield meaningful performance gains over the baseline choice of $S(s)=1$.
>
> As noted in lines 290-293, we are actively working to extend our framework to a safe Goal-Conditioned RL setting, where safety constraints are explicitly enforced. In such a context, a state-dependent speed term $S(s)$ is expected to play a more significant role in safety-aware value function learning and influence downstream evaluation metrics. We look forward to exploring this direction in future work.
>
> ## Q3
> > I suppose the weird behaviour of HIQL on the Figure 1 is because it is not fully trained yet and the value function resembles expected, optimal value function at the end. How can authors guarantee that the effect observed in Figure 1 is not a consequence of suboptimal hyperparameters?
>
> Thank you for the question. We would like to clarify that Fig. 1 is intended as an illustrative example to demonstrate the practical effect of our Eikonal regularizer on the learned value function.
> In the specific case of *antmaze-giant-navigate-v0* (shown in Fig. 1), we observe that with longer training, as reported in Table 2, the performance gap between Pi-HIQL and HIQL becomes negligible. However, this is not the case for all environments included in Table 2.
>
> The key takeaway is that while extended training may sometimes reduce the performance difference, the regularizer consistently introduces a useful inductive bias that improves sample efficiency and helps guide value learning early in training. This effect is particularly beneficial in settings with large environments, complex dynamics, and in the stitch regime.
>
> ## Q4, Q5, Q6
> > Training curves would be interesting to see to understand the Figure 1 in more details.
>
> We agree, and we have included all training curves in Appendix D, as mentioned in lines 321-323. The curve corresponding to Fig. 1, for the *antmaze-giant-navigate-v0* environment, is shown in Fig. 8.
>
> > It would also be interesting to consider the performance of the eikonal regularizer from the perspective of sample efficiency.
>
> If we define sample efficiency as the number of gradient steps required to reach satisfactory performance, our results suggest that the Eikonal regularizer improves efficiency. This is evident from the training curves in Appendix D. While the regularizer requires computing an additional term (Eq. 9) alongside the standard TD loss (Eq. 8), the computation is relatively efficient and adds only moderate overhead per training step. We believe this added complexity is justified by the performance gains observed across several benchmarks.
>
> > For the non-Lipschitz environments mentioned in the limitations. Does Eikonal regularizer improve the performance at the beginning of the training?
>
> As shown in the training curves in Appendix D, this is not the case. In highly non-Lipschitz environments, such as those involving contact-rich dynamics, we observe limited early-stage benefits. Addressing this limitation is an important direction for future work, and we are actively exploring ways to adapt our method to better handle such challenging settings.
>
> **References**
>
> [1] Kostrikov I, Nair A, Levine S. Offline reinforcement learning with implicit q-learning. arXiv preprint arXiv:2110.06169. 2021 Oct 12.
>
> [2] Xu H, Jiang L, Jianxiong L, Zhan X. A policy-guided imitation approach for offline reinforcement learning. Advances in Neural Information Processing Systems. 2022 Dec 6;35:4085-98.
>
> [3] Ghosh D, Bhateja CA, Levine S. Reinforcement learning from passive data via latent intentions. InInternational Conference on Machine Learning 2023 Jul 3 (pp. 11321-11339). PMLR.
>
> [4] Park S, Frans K, Eysenbach B, Levine S. Ogbench: Benchmarking offline goal-conditioned rl. arXiv preprint arXiv:2410.20092. 2024 Oct 26.

---

> ### Comment · Reviewer_zAXA · 2025-08-05
>
> I'd like to thank the authors for their rebuttal, it answered most of my concerns and with additional experiments I am happy to increase my score.
>
> One final remark I have is about the sample efficiency
>
> > If we define sample efficiency as the number of gradient steps required to reach satisfactory performance
>
> This is not the definition of sample efficiency I had in mind. Specifically I suggest the authors to look at situations, where collecting experiences in the enviroment is significantly more expensive than updating the model (for example robotic model trained directly in real world). In this situation samples can be iterated more frequently than they are collected, hence it does not correspond to gradient updates in a standard RL scenario. I recommend the authors to consider this situation a testbed for their work as i believe it can lead to potential benefits in this area.
>
> Thank you.

---

> > ### Author Response · Authors · 2025-08-05
> >
> > Thank you for your thoughtful feedback and the positive response.
> >
> > We appreciate your remark regarding sample efficiency. While our current work has been evaluated in an offline setting, we agree that environments where data collection is costly represent an important and relevant testbed.
> >
> > In this context, we believe the inductive bias introduced by our regularizer, which promotes faster convergence of the value function, has the potential to offer benefits in online scenarios as well, where sample efficiency is measured in terms of environment interactions. Investigating this effect in the online setting is indeed an interesting direction for future research.

---

### Note · Authors · 2025-08-11

We would like to sincerely thank all Reviewers and the Area Chair for their engagement, thoughtful questions, and constructive feedback throughout the review process. The discussion has been highly valuable in refining our work, and we are committed to incorporating the suggested improvements into our paper.

**Planned improvements**:

1. Include the additional experiments conducted with GCIQL and GCIVL to demonstrate that the Eikonal regularizer is applicable to other Temporal Difference-based GCRL methods beyond HIQL.

2. Clarify the terminology surrounding the Physics-informed regularizer in our approach.

3. Revise the phrasing in line 73 to more accurately reflect how model-based and Koopman operator frameworks leverage system dynamical properties.

4. Include the detailed derivations leading to the HJB equation in Eq. (5).

5. Provide a full step-by-step proof of Proposition 4.1 in the appendix.

6. Explicitly define the distance function $d(s)$ in Appendix B, add annotations to Fig. 5 to highlight the observed artifacts in *Pi-HIQL Exp* and *Pi-HIQL Lin*, and expand all acronyms upon first use in the introduction for improved clarity.

Thank you again for the valuable feedback and constructive discussion.

---

### Decision · Program_Chairs · 2025-09-17

**Decision:**

Accept (poster)

**Comment:**

(a) The paper introduces Pi HIQL, which extends HIQL for offline goal-conditioned RL. The method adds an Eikonal PDE–based regularizer to the value-learning objective. This regularizer enforces a distance-like structure by encouraging the gradient of the goal-conditioned value function (GCVF) to have constant magnitude. In practice, the authors augment HIQL’s expectile TD loss with an additive Eikonal penalty. This eikonal PDE prior is most conducive for domains with distance-like value/reward landscape (e.g. navigation). This formulation connects naturally to the hamilton jacobi bellman equation from optimal control and is a special case (more restrictive assumption). The resulting regularizer provides a model-free geometric prior that improves robustness and generalization when data coverage is limited. On OGBench, HIQL achieves large gains, especially in large mazes and stitching tasks. The improvements are more limited in contact-rich tasks where the value function is not globally smooth.

(b) The approach is a simple, grounded in PDEs and optimal control, and it is model-free and easy to integrate into TD-style learning. The theoretical motivation is clear, with a derivation from the HJB equation and a formal link to the Eikonal residual.
The empirical results are strong, with pronounced improvements on OGBench navigation and stitching benchmarks. Although it would have been good to characterize perfiormance and limitations on a broader range of more realistic problem domains (in semi contact rich environments).

(c) The scope of gains is mostly in navigation-dominated tasks, with limited benefits in contact-heavy domains like ant soccer and manipulation. The baseline coverage in the original submission was narrow, since the Eikonal regularizer was shown primarily with HIQL.
Although QRL and CRL were included as external baselines, there was no initial evidence that the regularizer helps other TD-based GCRL learners. The authors later committed to adding results with GCIQL and GCIVL. Some clarifications are missing in the submission, such as the exact definition of the distance function, annotations in figures, and fuller proofs and derivations. There is also a terminology issue: physics-informed is too broad, and reviewers suggested using Eikonal more explicitly.

(d) This is a technically sound paper. It delivers a low overhead inductive bias for offline GCRL. The paper shows clear wins where the hypothesis fits, i.e. smooth distance-like landscapes. The authors are also transparent about where it does not fit, namely contact-rich tasks. The connection to HJB is clear. The proposed loss is minimalistic, and the empirical gains in long-horizon navigation and stitching are both substantial and statistically meaningful.

(e) zAXA raised concerns about generality and evaluation scope, since the method was initially only shown with HIQL. The authors clarified that the regularizer is not tied to hierarchy or subgoals. Writing and clarity: XkBr asked for more precise terminology around Koopman/model-based phrasing and raised questions about sample efficiency. Reviewer 3kEX questioned parts of proposition 4.1 and the design choice of S(s) (resolved during rebuttal).